# Targeting Neurovascular Interaction in Retinal Disorders

**DOI:** 10.3390/ijms21041503

**Published:** 2020-02-22

**Authors:** Zhongjie Fu, Ye Sun, Bertan Cakir, Yohei Tomita, Shuo Huang, Zhongxiao Wang, Chi-Hsiu Liu, Steve S. Cho, William Britton, Timothy S. Kern, David A. Antonetti, Ann Hellström, Lois E.H. Smith

**Affiliations:** 1Department of Ophthalmology, Boston Children’s Hospital, Harvard Medical School, Boston, MA 02115, USA; zhongjie.fu@childrens.harvard.edu (Z.F.); Ye.Sun@childrens.harvard.edu (Y.S.); Bertan.Cakir@childrens.harvard.edu (B.C.); Yohei.Tomita@childrens.harvard.edu (Y.T.); Shuo.Huang@childrens.harvard.edu (S.H.); Zhongxiao.wang@childrens.harvard.edu (Z.W.); Chi-Hsiu.Liu@childrens.harvard.edu (C.-H.L.); Steve.Cho@childrens.harvard.edu (S.S.C.); William.britton@childrens.harvard.edu (W.B.); 2Manton Center for Orphan Disease, Boston Children’s Hospital, Boston, MA 02115, USA; 3Center for Translational Vision Research, Gavin Herbert Eye Institute, Irvine, CA 92697, USA; kernt@uci.edu; 4Kellogg Eye Center, Department of Ophthalmology and Visual Sciences, University of Michigan, Ann Arbor, MI 48105, USA; dantonet@med.umich.edu; 5Section for Ophthalmology, Department of Clinical Neuroscience, Institute of Neuroscience and Physiology, Sahlgrenska Academy, University of Gothenburg, 405 30 Göteborg, Sweden; ann.hellstrom@medfak.gu.se

**Keywords:** photoreceptors, energy shortage, inflammation, angiogenesis, retina

## Abstract

The tightly structured neural retina has a unique vascular network comprised of three interconnected plexuses in the inner retina (and choroid for outer retina), which provide oxygen and nutrients to neurons to maintain normal function. Clinical and experimental evidence suggests that neuronal metabolic needs control both normal retinal vascular development and pathological aberrant vascular growth. Particularly, photoreceptors, with the highest density of mitochondria in the body, regulate retinal vascular development by modulating angiogenic and inflammatory factors. Photoreceptor metabolic dysfunction, oxidative stress, and inflammation may cause adaptive but ultimately pathological retinal vascular responses, leading to blindness. Here we focus on the factors involved in neurovascular interactions, which are potential therapeutic targets to decrease energy demand and/or to increase energy production for neovascular retinal disorders.

## 1. Retinal Neurovascular Development

Retinal neurovascular crosstalk increases with the development of photoreceptors. As birth date studies of retinal neurogenesis in monkeys are relevant to human retinal development, human retinal development is referenced to that in monkeys. Neural development starts from the central retina and extends to the periphery [1]. In both human and monkey, the fovea, a region with high concentrations of cone photoreceptors, develops at 26–30% of full gestation while cone outer segments develop by 63–65% of full gestation [2]. Around the end of the first trimester (~30% gestation), the hyaloid artery (branched from the primitive dorsal ophthalmic artery), which extends through the primitive vitreous and provides nutrients to the developing eye [3], begins to regress, and in the following four months (~30 to 70% gestation), as the retinal vascular network extends from the central to the peripheral retina, the development of photoreceptors takes place [1]. The formation of the retinal vasculature is completed shortly after birth. In general, retinal angiogenesis begins in the superficial layers at the nerve fiber layer/ganglion cell layer interface. The first superficial primary capillary bed next sends branches into the retina to form the deep capillary bed, a new layer of vessels at the border of the INL/outer plexiform layer. The intermediate layer forms last in the inner plexiform layer and inner nuclear layer (INL). (Figure 1). The timing and spatial arrangement of the retinal vasculature and retinal neurons suggest a neurovascular link. However, it is important to note that the signals involved in the neurovascular interaction and their impact might be different between developmental and pathological conditions. For example, retinal glia cell-derived vascular endothelial growth factor (VEGF) is important for physiological retinal angiogenesis, but high VEGF expression in ischemic retinopathy induces pathological angiogenesis [4]. In addition, loss of WNT signaling leads to delayed hyaloid vessel regression and delayed retinal vascular growth, but over activation of WNT signaling leads to retinal neovascularization [5]. During the developmental stage, signals are tightly controlled for the proper formation of retinal vasculature. During disease conditions, the signals are uncontrolled and result in retinal vessel overgrowth.

The endothelial cell Norrin/Frizzled-4 pathway plays a key role in regulating retinal angiogenesis [6,7]. The formation of retinal vascular patterns is also tightly regulated by factors from the surrounding environment. For example, endothelial stalk cells proliferate in response to a gradient of tissue-hypoxia-induced VEGF [8]. Retinal ganglion cell GPR91 responds to increased succinate levels and induces retinal vascular sprouting in hypoxic rodent retinas [9]. We will discuss the inductive signals and guidance cues involved in the neurovascular interaction and their contribution to the development of retinal disorders. 

## 2. Factors Influence Retinal Neurovascular Interaction

Retinal vasculature is critical for the preservation of normal retinal function. This is because most nutrients (oxygen, glucose, lipids, ions) delivered to neurons of the inner retina come from the retinal vasculature, with the majority of photoreceptor cell nutrients provided by the choriocapillaris, which lies just below the retinal pigmentary epithelium (RPE). 

Retinal photoreceptor cells are critical to the normal function of the retina, and play a major role in the neurovascular unit. Photoreceptor cell signaling is implicated in the degeneration of retinal capillaries in a variety of diseases, including DR, genetic diseases that lead to photoreceptor degeneration, and also in retinopathy of prematurity (ROP)**.** It has been postulated that oxygen demands of the photoreceptors cause the development of the vascular abnormalities that are characteristic of ROP [10,11]. In photoreceptor-degenerating mice exposed to oxygen in oxygen-induced retinopathy (OIR, modeling ROP), there are fewer preretinal vascular endothelial cell nuclei and reduced retinal VEGF expression level versus wild type (WT) OIR controls [12]. Therefore, understanding the roles of photoreceptor metabolic needs and factors involved in regulating retinal vascular responses is critical for disease prevention and treatment. In addition, retinal ganglion cells (RGCs) play an important role in maintaining the normal structure and function of retinal blood vessels. Previous studies show that retinal vascular networks fail to develop in mice lacking RGCs [9]. RGCs also contribute to retinal blood vessels in experimental models of retinal regeneration induced by ischemia–reperfusion [13,14]. These findings strongly suggest that RGCs play an important functional role in both physiological and pathological retinal angiogenesis [15,16].

### 2.1. Oxygen Shortage and Retinal Neovascularization

The retinal vasculature provides oxygen and nutrients to support neurons. Oxygen is required for oxidative phosphorylation for mitochondrial energy production, in addition to lipid β-oxidation. In the light, retinal oxygen tension is highest in the choroid and decreases steeply between the choriocapillaris and the outer plexiform layer [17]. Photoreceptor inner segments have abundant mitochondria and consume the most oxygen in the retina [18,19]. It has also been reported that purified bovine rod outer segment, although devoid of mitochondria, have ectopically located mitochondrial proteins (respiratory chain complexes I-V) present, consume oxygen, and synthesize ATP [20,21,22]. In isolated bovine rod outer segment following photoexcitation, glycolysis alone is not sufficient to provide enough energy for phototransduction [23]. Phosphocreatine shuttle transports high energy phosphate groups in the form of creatine phosphate from rod inner segment to outer segment for conversion to ATP [23]. The pentose phosphate pathway also contributes to NADPH production [23]. However, in retinal metabolic disorders, oxygen saturation is altered [24]. In retinitis pigmentosa photoreceptors are lost and retinal oxygen consumption is reduced. This is correlated with visual field damage and retinal atrophy. In diabetic retinopathy, venous oxygen saturation increases and arteriovenous differences decrease. The magnitude of this difference correlates with disease severity. 

Most cells respond to hypoxia by increasing the stability of hypoxia-inducible factor 1 protein (HIF-1) [25], which in turn induces the transcription of HIF-1-dependent VEGF-A [26]. VEGF-A is critical for blood vessel and neural growth and stability [27,28]. In rodent models of proliferative retinopathy, relative hypoxia induces high levels of VEGF-A production, causing uncontrolled vessel growth [29,30]. Photoreceptor metabolic needs regulate this process [10,31]. In addition, lactate, produced with anaerobic glycolysis, stabilizes N-myc downstream-regulated genes, to induce the Raf-ERK pathway to promote angiogenesis in tumors via an HIF-independent pathway [32]. Overall, oxygen is essential for the maintenance of retinal cell function and cells respond to hypoxia by inducing vessel growth to compensate for oxygen shortage. 

### 2.2. Energy Shortage Drives Retinal Neovascularization

To produce energy for cells, fuel is needed. Previously, it was common knowledge that glucose was the only energy source for the retina [33]. However, we found that lipids are also used as mitochondrial fuel (oxidative phosphorylation) [34] accounting for the energy gap noted in 1960 that most glucose (~65%) is used for aerobic glycolysis rather than oxidative phosphorylation [35]. Thus, aberrant mitochondrial and peroxisomal fatty acid β-oxidation may lead to an energy shortage.

It is known that mutations in fatty acid transport proteins lead to retinal degeneration [36]. Studies have shown that very-low-density-lipoprotein receptor (VLDLR) loss-of-function in photoreceptors decreases retinal lipid uptake and results in accumulation of extracellular lipids. This in turn inhibits retinal glucose transporter 1 activity [34]. Ultimately, the resulting mitochondrial fuel shortage drives retinal neovascularization [34]. In addition, gene mutations in peroxisomal lipid transporters and metabolic enzymes may lead to retinal abnormalities, including pigmentary retinopathy and optic atrophy [37]. Mutations in Acyl-CoA binding domains containing protein 5 (which preferentially binds very-long-chain fatty acyl-CoAs) is associated with retinal dystrophy [38]. Deficiency of acyl-CoA binding domain containing protein 5 in HeLa cells leads to impaired peroxisomal β-oxidation and accumulation of very long-chain fatty acids [39]. However, knowledge of peroxisome function in retinal lipid metabolism is limited.

Energy shortage promotes pathological vascular growth. A fuel shortage of both lipid and glucose in photoreceptors of VLDLR knockout mice leads to a reduction in the levels of the Krebs cycle intermediate α-ketoglutarate. Low α-ketoglutarate levels (as well as low oxygen levels) stabilize HIF-1α protein and increase the production of VEGF-A, which in turn induces pathological vessel formation [34]. In the neonatal mouse model of hyperglycemia-associated retinopathy, decreased photoreceptor glucose metabolism causes a reduction in the photoreceptor platelet-derived growth factor beta expression, thereby delaying normal retinal vascular development [40]. Ketone bodies, mainly beta hydroxybutyrate and acetoacetate, are alternative energy sources in the fasting state [41]. Beta hydroxybutyrate activates G protein-coupled receptors GPR109A, which exhibits anti-inflammatory and neuroprotective effects [42]. Acetoacetate, as an endogenous agonist for GPR43, regulates energy expenditure and lipid metabolism in mice under fasting conditions [43]. Ketogenic diets are neuroprotective in some neuronal disorders, including ischemic stroke [42], Parkinson’s disease [44], and Alzheimer’s disease [45]. Exogenous ketone bodies also show neuroprotective effects on retinas with enhanced antioxidative defenses [46]. RPE cells in vitro have high expression of key enzymes involved in ketogenesis and phagocytose photoreceptor outer segment to produce ketone bodies [47,48]. RPE cells also metabolize ketone bodies for energy production [47]. High levels of ketone bodies are observed in patients with type 2 diabetes [49,50]. Insulin treatment elevates serum ketone bodies in type 1 diabetes [51]. A recent conference report shows that beta hydroxybutyrate via GPR109A reduces NLRP3-derived inflammation and loss of GPR109A worsens retinal vascular pathology in streptozotocin-induced type 1 diabetic retinas [52]. In general, there might be a protective role of ketone bodies in diabetic retinopathy but the impacts of ketone bodies on diabetic retina need to be further investigated. Taken together, further exploration in the interaction between photoreceptor energy deficiency and vascular development is needed for disease prevention and treatment at the early stages. 

### 2.3. Oxidative Stress and Retinal Neovascularization

Oxidative stress results from an imbalance between the antioxidant defense system and the production of reactive oxygen species (ROS). Phototransduction, oxidization of polyunsaturated acids, and phagocytosis of photoreceptor outer segment leads to chronic production of ROS, resulting in potential oxidative stress [53]. Oxygen-consuming mitochondria in the inner segments play a primary role in oxidative stress in the outer retina [54]. Recent studies also report that oxidative stress may occur directly in the photoreceptor OS after blue light irradiation [55]. In the purified bovine rod outer segment, a dose response is observed to varying light intensity and duration in terms of both reactive oxygen intermediates and ATP synthesis [20]. In normal conditions, retinal cells maintain homeostasis between pro- and anti-oxidative signaling [56]. For example, in the mitochondria, superoxide dismutase 2 converts the major form of ROS in living cells, superoxide radicals (O_2_^–^), into H_2_O_2_ and O_2_ [57]. In the peroxisome, oxidases reduce O_2_ to H_2_O_2_ during lipid breakdown, while catalase removes H_2_O_2_ [56,58]. However, when the antioxidant defense system is compromised, excessive ROS lead to damaged proteins, nucleic acids, and lipids, contributing to neuronal loss in many retinal diseases [59,60]. Increasing anti-oxidative signaling prevents neuronal loss in photoreceptor-degenerating mouse mutants [56], as well as in the mouse model of ischemic retina [61]. Oxidative stress is a significant risk factor for age-related macular degeneration (AMD). Diet poor in antioxidant micronutrients (vitamin C, E, carotenoids, zinc) and low plasma levels of antioxidants may favor the development of AMD [62]. Inhibition of oxidative stress with N-acetylcysteine attenuates spermidine-induced hyperpermeability of the blood-retinal barrier, decreased rod function, as well as RPE degeneration in rats [63]. In hyperglycemic states, metabolic pathways producing ROS are activated which enhance inflammatory, apoptotic, and degeneration pathways, ultimately leading to the development of diabetic retinopathy [64,65]. Therefore, increasing anti-oxidant defenses under conditions of oxidative stress may help treat retinal metabolic disorders.

### 2.4. Retinal Circuitry and Retinal Remodeling

Progressive photoreceptor (especially cone) loss in retinal degenerative diseases, such as retinal detachment, AMD, and retinitis pigmentosa, leads to extensive, phased changes in the remnant retinal circuitry—termed “retinal remodeling,” which is unavoidable at the cellular and molecular level in the inner retinal neurons and glial cells [66,67]. First, photoreceptor stress and metabolic alterations in glia are initiated; second, glial remodeling occurs with the loss of photoreceptors; third, there is a neural, glial, and vascular remodeling of the surviving retina. In retinitis pigmentosa patients, the severity of visual field loss is correlated with retinal vessel attenuation [68,69]. In retinitis pigmentosa animal models, progressive loss of retinal blood vessels is also observed and a significant decrease in capillary density and capillary loop is found particularly in the deep (close to photoreceptors), but not in the superficial and intermediate capillary plexus [70,71]. In ROP mouse models, hyperoxia causes reduced density of retinal astrocytes in the avascular zone and maintaining retinal astrocytes normalizes revascularization [72,73]. Migrating astrocytes associate closely with the axons of retinal ganglion cells and subsequently direct vessel development in mice [74]. However, the thickness changes are only observed in the inner nuclear and plexiform layer but not photoreceptors in the mouse OIR model at the time when maximum neovascularization occurs [75]. Therefore, photoreceptor stress-induced retinal remodeling may also control retinal vascular patterning possibly through modulating glial responses.

### 2.5. Inflammation and Retinal Neovascularization

Immune response and inflammation are associated with pathological angiogenesis as seen in ROP, AMD, and diabetic retinopathy [76,77,78,79,80]. An immature immune system may increase the risk for ROP in premature neonates [81]. Recent work suggests a strong link between the suppressor of cytokine signaling-3 (SOCS3) signaling pathway and ROP [82,83,84]. SOCS proteins are key regulators of both innate and adaptive immunity and inflammatory responses, and can positively and negatively regulate macrophage and dendritic-cell activation, respectively, and is essential for T-cell development and differentiation. SOCS proteins may be involved in diseases of the immune system in ocular neovascularization [85].

Toll-like receptors [86] and the transcriptional activator nuclear factor NF-ĸB [87] can potentiate inflammatory responses and promote further endothelial injury, resulting in generalized systemic inflammation [88]. Prolonged supplemental oxygen exposure as well as systemic inflammation markers, such as plasma interleukin 6 (IL6) and tumor necrosis factor-alpha (TNFα) are strongly associated with severe ROP [89]. Markers of inflammation appear to be associated with ROP in human studies [90]. Inflammatory processes might interfere with normal retinal vascularization in preterm retinas [91]. Suppression of TNFα in the mouse OIR model appears to be protective [92]. Omega-3-polyunsaturated fatty acids in OIR decrease the size of the avascular area and this protective effect may be mediated, in part, via suppression of TNFα [93]. Besides the OIR model, the expression levels of inflammatory cytokine (IL6 and TNFα) modulated by key inflammatory factor c-Fos in photoreceptors are increased in VLDLR knockout mice [94].

Immune dysfunction and inflammation changes has been clinically and experimentally linked to development of choroidal neovascularization (CNV) [95,96,97,98,99,100,101,102], which affects ~10% of AMD patients, but it accounts for up to 90% of vision loss associated with AMD [103]. Immune cells play a key role in linking innate and adaptive immunity, primarily through antigen presentation and recruitment of adaptive immune cells. Choroidal circulation brings to the eye a large number of immune cells, specifically those of myeloid origin [104,105,106], which play important roles in retinal and choroidal vascular pathology [100,101,107,108,109,110]. In response to chronic insults, such as those occurring in AMD, myeloid cells become activated and release inflammatory mediators (such as VEGF, matrix metalloproteinases, interleukins, and chemokines) that stimulate ocular NV [109,111,112]. A number of chemokines attract immune cells to invade and infiltrate ocular tissues, and furthermore, stimulate the secretion of more trafficking molecules that influence immune cell migration [113]. Therefore, inflammatory mediators and the degree of myeloid cell activation and infiltration around CNV is critical to the innate inflammatory response that contributes to CNV onset and progression. 

Development of pathological neovascularization in vascular eye diseases is linked with altered inflammation and macrophage function/polarization [114]. Lipid-induced alteration in inflammatory response is mediated in part by macrophages, which are phenotypically plastic with different polarization states (M1 and M2-like), and play critical roles in regulating ocular angiogenesis during development and in disease [115]. Retinoic-acid-receptor-related orphan receptor alpha (RORα), a lipid sensing nuclear receptor and transcription factor [116], which regulates lipid homeostasis and inflammatory cell differentiation and cytokine production [117], controls pathological angiogenesis through direct transcriptional control of SOCS3. This suggests the importance of lipid metabolism driven inflammation in ocular angiogenesis [84]. In addition, retinal gliosis had been reported to associate with ocular neovascularization, such as ROP [118] and diabetic retinopathy [119].

### 2.6. Neuron-Derived Factors and Neovascularization

Dysregulated crosstalk between the vasculature and retinal neurons, modulated by neuron-derived growth factors and guidance cues, contributes to the pathogenesis of ocular vascular diseases [15,120,121], stroke [122], Alzheimer’s disease [123], and epilepsy [124]. Among those neuron-derived factors, semaphorins (SEMAs), also known as collapsins, are critical regulators of angiogenesis during development and in diseases with neovascularization such as cancer and retinopathy. Semaphorins were first identified as a family of genes encoding guidance molecules for the embryologic development of the nervous system [125,126]. The action mechanisms underlying semaphorins through their receptors from different cell types in different disease conditions had been summarized previously [127,128,129]. Cerani et al. reported that neuron-derived Semaphorin3A is an early inducer of vascular permeability in diabetic retinopathy via neuropilin-1 [130]. Yang et al. reported that Semaphorin-3C signals through Neuropilin-1 and PlexinD1 receptors to inhibit pathological angiogenesis [131]. Wu et al. reported that inhibition of Sema4D and its receptor Plexin B1 signaling alleviates vascular dysfunction in diabetic retinopathy [132]. Fukushima et al. reported that Sema3E-PlexinD1 signaling selectively suppresses disoriented angiogenesis in ischemic retinopathy in mice [133]. These findings demonstrate that semaphorin family proteins are involved in neovascularization.

## 3. Therapeutic Potentials of Manipulating Pathways Controlling Neurovascular Crosstalk

Anti-VEGF agents are the standard of care to treat most pathological retinal vessel growth in eye diseases [134,135]. However, anti-VEGF treatment is not completely effective and is associated with some adverse effects [136,137,138]. Systemic inhibition of VEGF levels following intravitreal anti-VEGF administration in patients with proliferative retinopathies is particularly concerning in preterm infants with ROP and incomplete vascular development of many organs [139,140,141,142,143,144], as VEGF is an important factor to maintain normal neuronal and vascular function. Therefore, there is a need for new safe therapeutic approaches. Modulation of retinal metabolism and inflammatory factors might prevent or treat neurovascular retinal diseases (Figure 2). One approach might be to increase fatty acid β oxidation with hormonal modulation (peroxisome proliferator-activated receptor alpha (PPARα) agonists such as fenofibrate, long acting fibroblast growth factor 21 (FGF21), or adiponectin). Inhibition of the visual cycle (to decrease the energy demand), near infrared (IR) light photobiomodulation (to increase ATP production), as well as inflammatory regulation (c-Fos, SOCS3) may potentially improve photoreceptor function as a therapeutic approach. There are also other potential factors involved in neurovascular interaction. For example, abnormal expression of numerous retinal miRNAs is related to retinal disorders such as AMD, diabetic retinopathy, retinitis pigmentosa, and retinoblastoma in both human and animal models [145]. In addition, miRNAs also regulates angiogenic factors and controls angiogenesis in vitro and in vivo [146,147,148,149]. In the current review, we focus on the major signals regulating oxygen and nutrients that mostly related to retinal metabolism, as well as some of the potential therapeutic targets that have been investigated in our group.

### 3.1. Hormonal Modulation

Fenofibrate, which is a PPARα agonist essential for lipid use and cytochrome P450 2C antagonist [150], prevents the progression of DR. In type 2 diabetic patients, fenofibrate treatment reduces the risk of proliferative DR by 35–40% [151,152]. In a rat model, fenofibrate reduces retinal neovascularization and ameliorates retinal vascular leakage in murine diabetes [153]. Inhibition of cytochrome P450 2C with fenofibrate or other inhibitors such as Montelukast decreases pathologic retinal angiogenesis in mice [154,155]. PPARα may also function through inhibition of the Wnt/Norrin pathway [156]. Wnt proteins and Norrin are ligands of receptors such as low-density lipoprotein receptor-related protein 5 or 6 (LRP5/6) and frizzled 4, which leads to the stabilization of β-catenin, which then translocates into the nucleus where it binds to transcription factors of lymphoid enhancer factor/T-cell factor to activate transcription of Wnt target genes [157,158]. The Wnt/Norrin pathway plays a crucial role in vascular development and retinal neuronal survival [5,159]. However, over-activated Wnt signaling is associated with pathologic retinal angiogenesis [160]. Therefore, inhibitors targeting aberrant Wnt/Norrin signaling may be benefit ocular neurovascular disorders. 

Fasting or administration of PPARα agonist (such as fenofibrate) produces fibroblast growth factor (FGF21) in the liver [161,162,163,164]. FGF21, a secreted protein of 209 amino acids, was first reported in 2000 [165]. FGF21 administration lowers blood glucose levels and improves lipid metabolism in diabetic mice [166,167,168,169,170,171,172,173,174]. Clinically, FGF21 decreases body weight, decreases triglyceride levels and increases high-density lipoprotein cholesterol levels in type 2 diabetic patients [175,176]. In animal models of neurovascular eye disorders, including OIR, VLDLR knockout mice, and laser-induced choroidal neovascularization, FGF21 suppresses neovascularization in the retina and choroid [177], and protects against retinal dysfunction in the neural retina in diabetic mice [178], as well as decreasing blood-brain barrier leakage in type 2 diabetes mice and ameliorates transendothelial permeability in vitro [179].

Adiponectin is a circulating adipokine that is upregulated by FGF21 in adipose tissue [169,180,181]. Adiponectin is a key modulator of glucose and lipid metabolism, and is involved in many retinal metabolic disorders [182]. Circulating adiponectin levels are associated with the progression of retinal diseases such as ROP, DR, and AMD [183,184,185,186]. The depletion of *AdipoR1*, which is a major adiponectin receptor gene, causes a shortage of essential ω-3 very-long-chain polyunsaturated fatty acids in the retina and in turn photoreceptor degeneration in mice [186,187]. Modulating photoreceptor metabolism with adiponectin administration protects against retinal dysfunction and improves retinal vascular development in mice modeling early ROP [40]. The adiponectin pathway inhibits neovascularization in a number of animal models of proliferative retinopathy [183,188,189,190] and endothelial cell tube formation in vitro [191]. Adiponectin also dilates retinal arterioles via production of nitric oxide from endothelial cells [192]. The concentration of serum adiponectin is associated with increased retinal blood flow in type 2 diabetes patients [185]. Therefore, adiponectin is a potential therapeutic target in retinal metabolic diseases.

Adenosine monophosphate-activated protein kinase (AMPK), a crucial cellular energy sensor, is activated by falling energy status [193]. Activation of AMPK via attenuating NF-κB activation protects retinal neurons against lipopolysaccharide-induced inflammation [194]. Activation of AMPK with metformin protects light-induced retinal degeneration with decreased oxidative stress and DNA damage, as well as increased mitochondrial energy production [195]. Peroxisome proliferator-activated receptor coactivator-1alpha (PGC-1α), which is stimulated by metabolic sensor AMPK and sirtuin 1 (SIRT1), is a transcriptional coactivator of many genes involved in energy modulation and mitochondrial biogenesis [196,197]. PGC-1α repression and mitochondrial dysfunction are observed in RPE derived from AMD donor eyes versus age-matched normal controls [198]. A high-fat-diet induces AMD-like abnormalities in RPE and retinal morphology in mice with low levels of PGC-1α [199]. PGC-1α increases oxidative phosphorylation and fatty acid oxidation in RPE in vitro [200], as well as protecting RPE cells of the aging retina against oxidative stress-induced degeneration in vivo [196]. Loss of PGC-1α in mice leads to rapid RPE dysfunction and promotes photoreceptor degeneration [201], as well as severely protecting against a deterioration in retinal morphology and function with toxic light exposure [202]. PGC-1α also induces the expression of VEGFA in retinal cells and PGC-1α deficiency reduces early retinal vascular outgrowth, capillary density, and number of arteries and veins as adults [203]. In a mouse model of ROP, PGC-1α expression is dramatically induced in the inner nuclear layer, and loss of PGC-1α inhibits retinal neovascularization by decreasing VEGFA levels [203]. Adiponectin induces AMPK activation by promoting the cytosolic localization of LKB1 and stimulating Ca^2+^ release from intracellular stores in muscle cells [204]. In hyperglycemia-associated ROP, adiponectin via AMPK increases photoreceptor metabolism and pro-angiogenic growth factor *Pdgfb* production, and in turn improves retinal vascular growth [40]. Endocrine FGF21 is also an AMPK activator either directly through FGFR1/β-klotho signaling or indirectly by stimulating the secretion of adiponectin and corticosteroids [205]. FGF21 controls mitochondrial function by activating the AMPK-SIRT1-PGC-1α pathway in adipocytes [206].

### 3.2. Photobiomodulation

Photobiomodulation, also known as near infrared therapy (NIR) or far-infrared therapy (FIR), is a light-based treatment with a wave length between 600–1000 nm from a laser or a light emitting diode (LED). Over the last decades there has been an increased interest in the use of NIR and FIR in clinical and pre-clinical research. Clinically, it has shown promise in aiding wound healing [207], neurological neck pain [208], chronic joint disorders [208], peripheral nerve recovery [209], stroke recovery [210], and traumatic brain injury [211]. Photobiomodulation has also gained interest for the treatment of retinal disorders, having had positive effects in several mouse retinopathy models such as in light induced injury [212,213], OIR [214,215], streptozotocin-induced diabetes [216], methanol induced degeneration [217], and a model of complement factor H related degeneration [218]. These promising results have led to clinical trials in humans with AMD. In 2008, 203 patients with beginning “dry” and advanced AMD “wet” were treated with NIR light (0,3 J/cm^2^, 760 nm) transconjunctivally to the macula twice per week for 2 weeks [219]. Vision improved significantly in wet and dry AMD and was maintained for 3–36 months after treatment. In another study 2016 patients with dry AMD (AREDS. 2–4) were treated with multiwavelength light (590, 670, 790 nm) through the pupil three times per week for 6 weeks [220], which led to significant improvement of visual acuity and contrast sensitivity as well as a decrease in drusen volume and thickness. No adverse events were seen in these two studies.

Many different molecular and cellular mechanisms for the neuroprotective effect of near infra-red light have been proposed. The most commonly accepted mechanism is that cytochrome c oxidase (or complex IV) of the mitochondrial respiratory chain absorbs wavelength in the red/near-infrared spectrum. The capability of cytochrome c oxidase to undergo oxidation and reduction is essential for the generation and maintenance of the mitochondrial transmembrane potential, which is critical for ATP production via oxidative phosphorylation [221] (Figure 3). A brief 670 nm exposure increases mitochondrial membrane potential [222], ATP production [221,223], and complex IV expression levels [221]. In vivo measurements in the retina with broadband near-infrared spectroscopy show a consistent progressive increase in cytochrome c oxidase 5 oxidation after 670 nm exposure for 5 minutes [224]. Photobiomodulation increases blood flow to neural tissue, increases nitric oxide levels, slows reactivation of the mitochondrial electron transport chain during reperfusion reduction in ROS levels, reduces nitric oxide levels, increases expression of cell survival mediators, and improves microglial function [225].

### 3.3. Suppression of the Visual Cycle to Decrease Energy Demands

Metabolism and function of photoreceptor cells and the retinal pigmentary epithelium show strong differences between day and night, and thus, diurnal differences on the outer retina can have important effects on the function of the retinal vasculature. The vertebrate visual system relies on two key pathways to detect and transform light signals into images; phototransduction and the visual (or retinoid) cycle. The high-energy-demanding phototransduction is vital for the conversion of external light into stimuli that can be processed by the brain to form an image, and is dependent on a constant supply of regenerated light-sensitive pigments. Evidence that neurovascular interactions, and specifically phototransduction, contribute to the vascular pathology at least in diabetic retinopathy can be derived from evidence that deletion of transducin, a key component of the phototransduction pathway, significantly inhibited the diabetes-induced degeneration of retinal capillaries and some molecular processes believed to participate in the pathogenesis of the retinopathy [227]. Likewise for the visual cycle, inhibition of a key enzyme of the visual cycle, RPE65, significantly inhibited both the diabetes-induced increase in retinal vascular permeability and retinal capillary degeneration over an 8 month period [228]. Administration of a visual cycle inhibitor (Emixustat) during the period of ischemia and reperfusion injury produced a ~30% reduction in retinal neovascularization in a mouse OIR model [229]. These observations suggest that signaling processes that are affected by the presence or absence of light can affect the integrity of the retinal neurovascular unit, and specifically the vasculature. Many more studies will be required to better understand the mechanisms and significance of these findings.

### 3.4. Inflammatory Regulation

Recently, there have been attempts to suppress neovascularization through controlling key regulators of inflammatory signals in the eye. Inflammatory changes in photoreceptors influence pathological angiogenesis in DR [230,231,232,233] and photoreceptor c-Fos controls proliferative angiogenesis by modulating photoreceptor inflammatory signals [94]. 

The expression levels of c-Fos and inflammatory cytokine (IL6 and TNFα) in photoreceptors are increased in VLDLR knockout mouse retinas, a retinal angiogenesis mouse model [94]. The immediate early gene and pro-oncogene c-Fos encodes a nuclear phosphoprotein that forms heterodimeric complexes with the Jun-family proteins to constitute activator protein 1 complex [234,235], which regulates inflammatory signals [236] during inflammatory disease development [237,238]. c-Fos is present in human photoreceptor cells throughout development [239] and regulates rod-specific gene expression [240] and photoreceptor apoptosis [235,241,242,243], and strongly associates with metabolic demands [94,244]. c-Fos is induced throughout the process of neovascularization formation in photoreceptors and controls retinal angiogenesis by modulating inflammatory signals in a retinal angiogenesis model [94]. Therefore, the c-Fos pathway may be a novel target to control inflammatory signals in stressed photoreceptors to mediate the development of neovascularization in ocular vascular diseases.

In addition, SOCS proteins are key physiological regulators of both innate and adaptive immunity and inflammatory responses. SOCS3 positively and negatively regulates macrophage and dendritic-cell activation, respectively, and it is essential for T-cell development and differentiation. SOCS proteins are involved in diseases of the immune system [85]. SOCS3 is also required to temporally fine-tune photoreceptor cell differentiation [245]. Therefore, the SOCS3 pathway might also play a role in regulating photoreceptor inflammatory signals in ocular neovascularization. 

### 3.5. Class 3 SEMAs

Class 3 SEMAs consist of seven secreted soluble proteins of ∼100 kDa in size (designated by the letters a-g), which are secreted by cells of multiple lineages, including epithelial cells, neurons, and specific tumor cells [126]. Class 3 SEMAs regulate cell–cell interactions, cell differentiation, morphology, and function [246]. SEMA3A, 3E, and 3F are vaso-repulsive cues in developmental retinal angiogenesis [247,248,249]. SEMA3A, 3C, and 3E are all involved in pathological retinal neovascularization in eye disease models [131,133,250]. Both SEMA3A and 3E are expressed in retinal ganglion cells [16,133]. SEMA3F is expressed in the outer retina while Sema3a is predominantly induced in the inner retina under hypoxic conditions [248]. SEMAs and their neuropilin and plexin receptors can modulate angiogenesis [133,250,251]. Inhibition of SEMA3E–plexin-D1 signaling results in an increase of pathological neovessels, whereas activation of SEMA3E–plexin-D1 signaling suppresses the outgrowth of neovessels in the mouse model of OIR [133]. However, the upstream factors that mediate the expression of SEMA3s in the retinas to influence neurovascular interaction and pathological retinal angiogenesis are largely unknown. RORα is a critical upstream regulator of SEMA3E expression, and controls SEM3E-dependent neurovascular coupling in retinopathy [16]. RORα directly binds to a specific ROR response element on the promoter of *Sema3e* and negatively regulates *Sema3e* transcription, which can impact retinopathy development via signaling through its receptor complexes located on pathological neovessels. Transcriptional regulation of the RORα-SEMA3E axis in the retinal ganglion cell layer mediates neurovascular crosstalk in pathological retinal angiogenesis. Taken together, therapies targeting SEMA3E are potentially promising approaches for the treatment of microvascular diseases, particularly microvascular retinopathies [246].

SEMA3F acts as an anti-angiogenic modulator suppressing the formation of pathological neovascularization in the outer retina [252]. Anti-angiogenic activity of SEMA3F in ocular neovascularization was found in VLDLR knockout mice and the mouse model of laser-induced choroidal neovascularization. SEMA3F is physiologically expressed in the retina of both mice and humans [248], suggesting that SEMA3F a promising target for treating pathological neovascularization. Intravitreal injection of anti-VEGF compounds is clinically well established [253,254,255] and combinatory treatments are currently under investigation in clinical studies [256]. Intravitreal treatment with SEMA3F alone or in combination with other compounds may be feasible for future treatment strategies in neovascular eye diseases.

## 4. Conclusions and Perspectives

Neuronal metabolism and inflammatory responses control retinal vascular function. Hormonal modulation and photobiomodulation to increase ATP production, or inhibition of the visual cycle may improve retinal function. Furthermore, controlling photoreceptor inflammatory responses by modulating c-Fos and SOCS3 pathways, as well as regulating neuron-derived Class 3 SEMAs to inhibit the pathologic vessel formation is also a promising field of study. To date, fenofibrate treatment in type 2 diabetic patients reduces proliferative diabetic retinopathy. Photobiomodulation in AMD patients also improves visual function. Long-acting FGF21 improves circulating lipid profiles and decreases body weight in type 2 diabetic patients. The investigation of FGF21 on retinal metabolic disorders are in the pre-clinical phase but clinical trials of long-acting FGF21 in patients with non-alcoholic steatohepatitis is ongoing. The development of safe and effective visual cycle inhibitors for retinal degeneration and proliferative retinopathies, and further exploration of modulating retinal inflammatory responses by c-Fos and SOCS3 in proliferative retinopathies are currently in process.

## Figures and Tables

**Figure 1 ijms-21-01503-f001:**
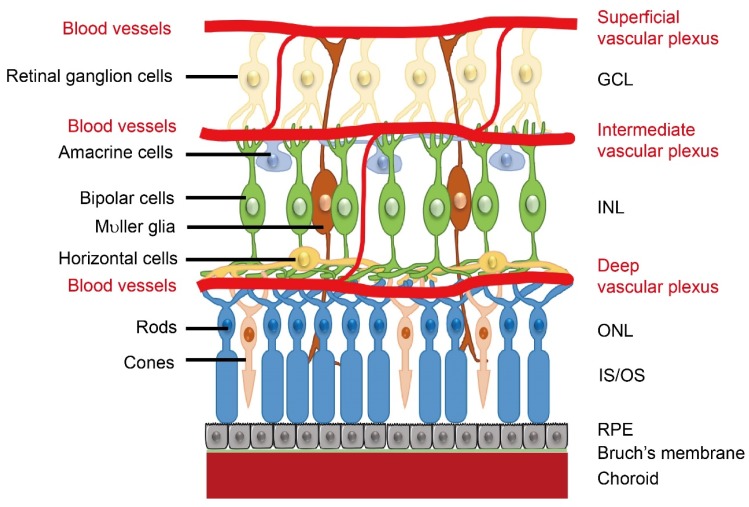
Schematics of retinal neuronal and vascular structure. There are three layers of retinal vascular plexuses tightly coordinated with retinal neurons. GCL, ganglion cell layer; INL, inner nuclear layer; ONL, outer nuclear layer; OS/IS, outer segments/inner segments; RPE, retinal pigment epithelium. Schematics were drawn by Dr. Shuo Huang, Ophthalmology, BCH.

**Figure 2 ijms-21-01503-f002:**
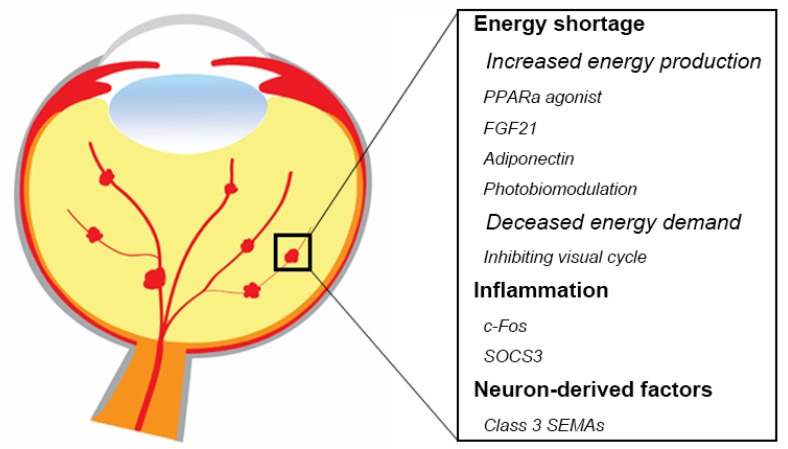
Summary of potential therapeutic targets in retinal neurovascular disorders. Modulating energy shortage, inflammation, as well as neuron-derived factors would be beneficial to protect against retinal neurons and inhibits pathological angiogenesis. Schematics were drawn by Dr. Shuo Huang, Ophthalmology, BCH.

**Figure 3 ijms-21-01503-f003:**
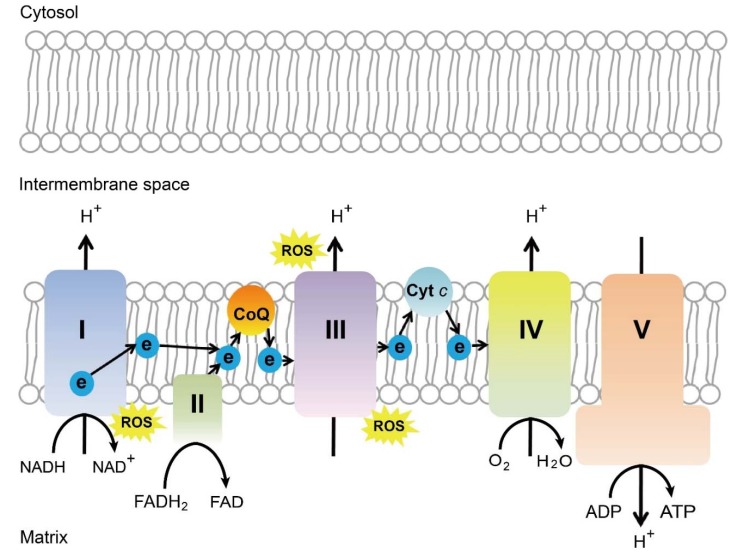
Schematics of mitochondrial electron transport chain. ROS, reactive oxygen species, ATP, adenosine triphosphate, CoQ, coenzyme Q, Cyt *c*, cytochrome *c*, Complex I, NADH coenzyme Q reductase, complex II, succinate dehydrogenase, complex III, cytochrome *bc_1_* complex, complex IV, cytochrome *c* oxidase, complex V, ATP synthase. Schematics were drawn by Dr. Zhongjie Fu, Ophthalmology, BCH. Adapted from EMBO Mol Med (2019)11: e10473. [226]

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
