# Peer review of "Targeting Neurovascular Interaction in Retinal Disorders"

_ijms, 2020, doi:10.3390/ijms21041503_

Round 1

Reviewer 1 Report

Dear Editor,

Aim of the paper from Fu et al. is to review the most important factors involved in neurovascular interactions, their role in the development of retinal disorders, and the potential therapeutic targets, with a very interesting focus on energy demand and production.  I have found this approach very interesting.  Quality of presentation article is written in an appropriate way. the scientific question is quite original and well defined, the study is scientifically sound, and its overall merit: is good. English language is sound.

Literature is cited correctly, apart form  recent literature data were not discussed in relation to the ultimate site of the elevated retinal oxygen consumption and of the origin of the oxidative stress in the outer retina. In 2020 the Authors cannot ignore the recent evidence for the presence of aerobic metabolism in the disks of the rod outer segments, that render these a primary source of reactive oxygen species, and oxidative stress, consistent with the idea that photoreceptors and not the RPE are the principal target of choroidal hypoxia and the main producer of oxidative stress, playing a central role in retinal degeneration.

The only schism is about the fact that this reviewer believes that oxidative stress caused by the extramitochondrial oxidative phosphorylation occurring inside the rod OS as reported (doi: 10.1016/j.biocel.2009.08.013. )

It is widely accepted that oxidative stress plays a central role in retinal degeneration: the uncertain issue is the actual source of it. 

For example, in AMD, the multiple risk factors (obesity,  smoking, and light exposure, and others) increase oxidative stress production.  the antioxidant N-acetylcysteine inhibited increasing oxidative stress caused by intravitreal injection of spermidine in the rat causes dry AMD. Hyperglycemia also was shown to affect epigenetic regulatory pathways, which  in turn affect the expression of genes related to oxidative stress in the development of DR. Edaravone, a strong antioxidant that prevents oxidative stress and a neuroprotective agent, indicated to slow down ALS progression was shown to protect against N-methyl-d-aspartate-induced retinal thinning by reducing oxidative stress.

 loss of photoreceptors is a prerequisite for axonal degeneration, and photoreceptor loss itself caused by oxidative stress production, initiates the chain of events that ultimately lead to retinal damage.

This is relevant if Authors are truly in search for the ultimate source of the oxidative stress that is clearly implicated in the development of the retinal diseases.  

Authors quite acritically state that : “Mathematic model predict that oxygen  consumption is highest in the photoreceptor inner segment where most photoreceptor mitochondria  are located [14]. “ in fact this paper is an old (1990) one in which the current opinion on photoreceptor use of anaerobic glycolysis was currently still on. Since 1994, Molday demonstrated that glycolysis is not sufficient to supply enough ATP for phototransduction in light.

Neurovascular interaction, are surely fundamental; retinal photoreceptor cells play an important role in the pathogenesis of retinal microvascular lesions in diabetes phototransduction, contribute to the vascular pathology at least in diabetic retinopathy:  deletion of transducing significantly inhibited the diabetes-induced degeneration of retinal capillaries (ref 148)

This reviewer has unpublished evidence that photo-biomodulation with NIR light does increase ATP production, by promoting the Cytochrome c Oxidase activity knot only in the mitochondria but especially in the rod OS. In fact the complex IV progressive increase in cytochrome c oxidase 5 oxidation after 670nm exposure for 5 270 minutes [146]. Photobiomodulation increases blood more expressed and likely accounting for the high oxygen consumption of the retina whose mitochondrial mass si not particularly elevated)

Authors themselves state that:” The high-energy-demanding phototransduction is vital for the conversion of external light into stimuli that can be processed by the brain to form an image, and is dependent on a constant supply of regenerated light-sensitive pigments. Evidence that neurovascular interactions, and specifically phototransduction, contribute to the vascular pathology”

In fact, photoreceptor loss reduces retinal oxygen consumption in retinitis pigmentosa, for example.

This reviewer is positive about the fact that “Retinal photoreceptor cells are critical to the normal function of the retina, and play a major role in the neurovascular unit...” Moreover, Retinal neurovascular crosstalk is critical in the photoreceptor development.

The extra-mitochondrial respiratory complexes present in the rod OS would provide energy to the phototransduction. This would explain the fact that deletion of transducin, inhibited the diabetes-induced degeneration of retinal capillaries and the pathogenesis of the retinopathy:  phototransduction causes an overwork of the electron transfer chain (see:  DOI: 10.1016/j.freeradbiomed.2018.01.029) the main ROS producer, in a site that is prone to lipid peroxidation, due to the large surface area of the disk membranes enriched with polyunsaturated lipids that would trigger free radical production causing MDA production.  

Authors should discuss and cite the papers showing that the mitochondrial redox chain proteins as well as ATP synthase are ectopically expressed, not only in the mitochondria but also in the rod Outer Segments along with the entire set of enzymes catalyzing oxidation of glucose and aerobic ATP synthesis (glycolysis, Krebs cycle, respiratory chain and ATP synthase).   And those from Roelke et al. showing that the oxidative stress is produced not only in the inner segment containing the mitochondria, but also in the Outer segments.

The work could provide a real advance from the current knowledge, if it considered a source of oxidative stress inside the photoreceptor outer segment (i.e.: the mitochondrial complexes of the respiratory chain ) This  data would offer a better explanation for all of the topics that are raised in the manuscript.

Author Response

Reviewer 1

Aim of the paper from Fu et al. is to review the most important factors involved in neurovascular interactions, their role in the development of retinal disorders, and the potential therapeutic targets, with a very interesting focus on energy demand and production.  I have found this approach very interesting. Quality of presentation article is written in an appropriate way. the scientific question is quite original and well defined, the study is scientifically sound, and its overall merit: is good. English language is sound.

Literature is cited correctly, apart from recent literature data were not discussed in relation to the ultimate site of the elevated retinal oxygen consumption and of the origin of the oxidative stress in the outer retina. In 2020 the Authors cannot ignore the recent evidence for the presence of aerobic metabolism in the disks of the rod outer segments, that render these a primary source of reactive oxygen species, and oxidative stress, consistent with the idea that photoreceptors and not the RPE are the principal target of choroidal hypoxia and the main producer of oxidative stress, playing a central role in retinal degeneration.

The only schism is about the fact that this reviewer believes that oxidative stress caused by the extramitochondrial oxidative phosphorylation occurring inside the rod OS as reported (doi: 10.1016/j.biocel.2009.08.013.) It is widely accepted that oxidative stress plays a central role in retinal degeneration: the uncertain issue is the actual source of it. For example, in AMD, the multiple risk factors (obesity, smoking, and light exposure, and others) increase oxidative stress production. The antioxidant N-acetylcysteine inhibited increasing oxidative stress caused by intravitreal injection of spermidine in the rat causes dry AMD. Hyperglycemia also was shown to affect epigenetic regulatory pathways, which in turn affect the expression of genes related to oxidative stress in the development of DR. Edaravone, a strong antioxidant that prevents oxidative stress and a neuroprotective agent, indicated to slow down ALS progression was shown to protect against N-methyl-d-aspartate-induced retinal thinning by reducing oxidative stress. Loss of photoreceptors is a prerequisite for axonal degeneration, and photoreceptor loss itself caused by oxidative stress production, initiates the chain of events that ultimately lead to retinal damage. This is relevant if Authors are truly in search for the ultimate source of the oxidative stress that is clearly implicated in the development of the retinal diseases.

Authors quite acratically state that: “Mathematic model predict that oxygen consumption is highest in the photoreceptor inner segment where most photoreceptor mitochondria are located [14]. “in fact this paper is an old (1990) one in which the current opinion on photoreceptor use of anaerobic glycolysis was currently still on. Since 1994, Molday demonstrated that glycolysis is not sufficient to supply enough ATP for phototransduction in light.

Neurovascular interaction, are surely fundamental; retinal photoreceptor cells play an important role in the pathogenesis of retinal microvascular lesions in diabetes phototransduction, contribute to the vascular pathology at least in diabetic retinopathy: deletion of transducing significantly inhibited the diabetes-induced degeneration of retinal capillaries (ref 148). This reviewer has unpublished evidence that photo-biomodulation with NIR light does increase ATP production, by promoting the Cytochrome c Oxidase activity not only in the mitochondria but especially in the rod OS. In fact the complex IV progressive increase in cytochrome c oxidase 5 oxidation after 670nm exposure for 5 minutes [146]. Photobiomodulation increases blood more expressed and likely accounting for the high oxygen consumption of the retina whose mitochondrial mass is not particularly elevated).

Authors themselves state that:” The high-energy-demanding phototransduction is vital for the conversion of external light into stimuli that can be processed by the brain to form an image, and is dependent on a constant supply of regenerated light-sensitive pigments. Evidence that neurovascular interactions, and specifically phototransduction, contribute to the vascular pathology”. In fact, photoreceptor loss reduces retinal oxygen consumption in retinitis pigmentosa, for example. This reviewer is positive about the fact that “Retinal photoreceptor cells are critical to the normal function of the retina, and play a major role in the neurovascular unit...” Moreover, Retinal neurovascular crosstalk is critical in the photoreceptor development. The extra-mitochondrial respiratory complexes present in the rod OS would provide energy to the phototransduction. This would explain the fact that deletion of transducin, inhibited the diabetes-induced degeneration of retinal capillaries and the pathogenesis of the retinopathy: phototransduction causes an overwork of the electron transfer chain (see: DOI: 10.1016/j.freeradbiomed.2018.01.029) the main ROS producer, in a site that is prone to lipid peroxidation, due to the large surface area of the disk membranes enriched with polyunsaturated lipids that would trigger free radical production causing MDA production.

Authors should discuss and cite the papers showing that the mitochondrial redox chain proteins as well as ATP synthase are ectopically expressed, not only in the mitochondria but also in the rod Outer Segments along with the entire set of enzymes catalyzing oxidation of glucose and aerobic ATP synthesis (glycolysis, Krebs cycle, respiratory chain and ATP synthase). And those from Roelke et al. showing that the oxidative stress is produced not only in the inner segment containing the mitochondria, but also in the Outer segments. The work could provide a real advance from the current knowledge, if it considered a source of oxidative stress inside the photoreceptor outer segment (i.e.: the mitochondrial complexes of the respiratory chain) This data would offer a better explanation for all of the topics that are raised in the manuscript.

Response: We thank the reviewer for the valuable suggestions. We have now included oxygen consumption and ATP synthesis in rod OS and rod OS as a potential source of oxidative stress in the revised manuscript, also shown below.

Photoreceptor inner segments have abundant mitochondria and consume most oxygen in the retina [1, 2]. It has also been reported that purified bovine rod OS, although devoid of mitochondria, have ectopically located mitochondrial proteins (respiratory chain complexes I-V) present, consume oxygen and synthesize ATP [3-5]. In isolated bovine rod OS following photoexcitation, glycolysis alone is not sufficient to provide enough energy for phototransduction [6]. Phosphocreatine shuttle transports high energy phosphate groups in the form of creatine phosphate from rod IS to OS for conversion to ATP [6]. The pentose phosphate pathway also contributes to NADPH production [6].

Oxidative stress results from an imbalance between the antioxidant defense system and the production of reactive oxygen species (ROS). Phototransduction, oxidization of polyunsaturated acids, and phagocytosis of photoreceptor outer segment leads to chronic production of ROS resulting in potential oxidative stress [7]. Oxygen-consuming mitochondria in the inner segments play a primary role in oxidative stress in the outer retina [8]. Recent studies also report that oxidative stress may occur directly in the photoreceptor OS after blue light irradiation [9]. In purified bovine rod OS, a dose response is observed to varying light intensity and duration in terms of both reactive oxygen intermediates and ATP synthesis [3]. In normal conditions, retinal cells maintain homeostasis between pro- and anti-oxidative signaling [10]. For example, in the mitochondria, superoxide dismutase 2 converts the major form of ROS in living cells, superoxide radicals (O2), into H2O2 and O2 [11]. In the peroxisome, oxidases reduce O2 to H2O2 during lipid breakdown while catalase removes H2O2 [10, 12]. However, when the antioxidant defense system is compromised, excessive ROS lead to damaged proteins, nucleic acids, and lipids, contributing to neuronal loss in many retinal diseases [13, 14]. Increasing anti-oxidative signaling prevents neuronal loss in photoreceptor-degenerating mouse mutants [10], as well as in the mouse model of ischemic retina [15]. Oxidative stress is a significant risk factor for age-related macular degeneration (AMD). Diet poor in antioxidant micronutrients (vitamin C, E, carotenoids, zinc) and low plasma levels of antioxidants may favor the development of AMD [16]. Inhibition of oxidative stress with N-acetylcysteine attenuates spermidine-induced hyperpermeability of the blood-retinal barrier, decreased rod function, as well as RPE degeneration in rats [17]. In hyperglycemic states, metabolic pathways producing ROS are activated which enhance inflammatory, apoptotic, and degeneration pathways, ultimately leading to the development of diabetic retinopathy [18, 19]. Therefore, increasing anti-oxidant defenses under conditions of oxidative stress may help treat retinal metabolic disorders.

  1. Hoang, Q. V.; Linsenmeier, R. A.; Chung, C. K.; Curcio, C. A., Photoreceptor inner segments in monkey and human retina: mitochondrial density, optics, and regional variation. Visual neuroscience 2002, 19, (4), 395-407.
  2. Yu, D. Y.; Cringle, S. J., Oxygen distribution and consumption within the retina in vascularised and avascular retinas and in animal models of retinal disease. Progress in retinal and eye research 2001, 20, (2), 175-208.
  3. Calzia, D.; Degan, P.; Caicci, F.; Bruschi, M.; Manni, L.; Ramenghi, L. A.; Candiano, G.; Traverso, C. E.; Panfoli, I., Modulation of the rod outer segment aerobic metabolism diminishes the production of radicals due to light absorption. Free radical biology & medicine 2018, 117, 110-118.
  4. Panfoli, I.; Calzia, D.; Bruschi, M.; Oneto, M.; Bianchini, P.; Ravera, S.; Petretto, A.; Diaspro, A.; Candiano, G., Functional expression of oxidative phosphorylation proteins in the rod outer segment disc. Cell Biochem Funct 2013, 31, (6), 532-8.
  5. Panfoli, I.; Calzia, D.; Bianchini, P.; Ravera, S.; Diaspro, A.; Candiano, G.; Bachi, A.; Monticone, M.; Aluigi, M. G.; Barabino, S.; Calabria, G.; Rolando, M.; Tacchetti, C.; Morelli, A.; Pepe, I. M., Evidence for aerobic metabolism in retinal rod outer segment disks. The international journal of biochemistry & cell biology 2009, 41, (12), 2555-65.
  6. Hsu, S. C.; Molday, R. S., Glucose metabolism in photoreceptor outer segments. Its role in phototransduction and in NADPH-requiring reactions. The Journal of biological chemistry 1994, 269, (27), 17954-9.
  7. Nishimura, Y.; Hara, H.; Kondo, M.; Hong, S.; Matsugi, T., Oxidative Stress in Retinal Diseases. Oxidative medicine and cellular longevity 2017, 2017, 4076518.
  8. Jarrett, S. G.; Lin, H.; Godley, B. F.; Boulton, M. E., Mitochondrial DNA damage and its potential role in retinal degeneration. Progress in retinal and eye research 2008, 27, (6), 596-607.
  9. Roehlecke, C.; Schumann, U.; Ader, M.; Brunssen, C.; Bramke, S.; Morawietz, H.; Funk, R. H., Stress reaction in outer segments of photoreceptors after blue light irradiation. PloS one 2013, 8, (9), e71570.
  10. Xiong, W.; MacColl Garfinkel, A. E.; Li, Y.; Benowitz, L. I.; Cepko, C. L., NRF2 promotes neuronal survival in neurodegeneration and acute nerve damage. The Journal of clinical investigation 2015, 125, (4), 1433-45.
  11. Fukui, M.; Zhu, B. T., Mitochondrial superoxide dismutase SOD2, but not cytosolic SOD1, plays a critical role in protection against glutamate-induced oxidative stress and cell death in HT22 neuronal cells. Free radical biology & medicine 2010, 48, (6), 821-30.
  12. Wanders, R. J.; Waterham, H. R., Biochemistry of mammalian peroxisomes revisited. Annual review of biochemistry 2006, 75, 295-332.
  13. Datta, S.; Cano, M.; Ebrahimi, K.; Wang, L.; Handa, J. T., The impact of oxidative stress and inflammation on RPE degeneration in non-neovascular AMD. Progress in retinal and eye research 2017, 60, 201-218.
  14. Calderon, G. D.; Juarez, O. H.; Hernandez, G. E.; Punzo, S. M.; De la Cruz, Z. D., Oxidative stress and diabetic retinopathy: development and treatment. Eye (Lond) 2017, 31, (8), 1122-1130.
  15. Li, S. Y.; Fu, Z. J.; Ma, H.; Jang, W. C.; So, K. F.; Wong, D.; Lo, A. C., Effect of lutein on retinal neurons and oxidative stress in a model of acute retinal ischemia/reperfusion. Investigative ophthalmology & visual science 2009, 50, (2), 836-43.
  16. Drobek-Slowik, M.; Karczewicz, D.; Safranow, K., [The potential role of oxidative stress in the pathogenesis of the age-related macular degeneration (AMD)]. Postepy Hig Med Dosw (Online) 2007, 61, 28-37.
  17. Ohashi, K.; Kageyama, M.; Shinomiya, K.; Fujita-Koyama, Y.; Hirai, S. I.; Katsuta, O.; Nakamura, M., Spermidine Oxidation-Mediated Degeneration of Retinal Pigment Epithelium in Rats. Oxidative medicine and cellular longevity 2017, 2017, 4128061.
  18. Cecilia, O. M.; Jose Alberto, C. G.; Jose, N. P.; Ernesto German, C. M.; Ana Karen, L. C.; Luis Miguel, R. P.; Ricardo Raul, R. R.; Adolfo Daniel, R. C., Oxidative Stress as the Main Target in Diabetic Retinopathy Pathophysiology. Journal of diabetes research 2019, 2019, 8562408.
  19. Kowluru, R. A.; Mishra, M., Oxidative stress, mitochondrial damage and diabetic retinopathy. Biochimica et biophysica acta 2015, 1852, (11), 2474-83.

Reviewer 2 Report

In the present review, the authors point to the role of the neurovascular unit in the retina focusing on the factors involved in neurovascular interactions as potential therapeutic targets to counteract neovascular retinal disorders.

Retinal neurons, their supporting cells and the vascular network work in close coordination to integrate vascular flow with retinal metabolic activity. As a result of correct relationships, an appropriate environment contributes to correct metabolic need that is required for correct visual function.

Here, particular emphasis has been given to the critical role played by retinal photoreceptors in the neurovascular unit. In particular, the vascular demands by photoreceptors in hypoxic condition as in ROP cause increased levels of HIF and VEGF that determine neovessel proliferation affecting in turn retinal circuitry and function. In RP, photoreceptor loss causes a reduction of oxygen consumption that leads to reduced levels of HIF and VEGF causing, in turn, vessel attenuation that participates to retinal dysfunction. Additional mechanisms underlying neovessel formation have been also detailed by the authors as responsible of retinal neovascularization such as oxidative stress, inflammation, neuron-derived factors, but the list can be continued. The same applies to the list of potential therapeutic targets in retinal neurovascular disorders. Therefore, either the authors provide a more extensive revision of the literature or make some effort to allow the reader to better understanding the basic principles underlying the takehome message of their review.  

In addition to include a more extensive revision on data on neurovascular network in the retina, the authors should also add some revision about the neuroretina and its dysfunction in relation to its interaction with the vascular network as also detailed in the title of the review.

Although much is known on dysfunctional vascular network in retinal disorders, a more detailed revision on alterations in neuroretina circuitry coupled to diseased retinal vasculature needs to be provided. For instance, ROP models are known to exhibit a reduced number of astrocytes in the retina with no significant changes in the number of neuronal cells and the thickness of retinal layers suggesting that the reduced number of astrocytes leads to an impairment of cellular interactions between glial cells and neuronal and/or vascular cells characterizing ROP. In rodent models of RP, in contrast, loss of photoreceptors leads to major remodeling of retinal circuitry that leads to changes in the physiology of downstream retinal ganglion cells. Much discussion is needed about the reciprocal contribution of retinal vessels and retinal circuitry that, together, participate to a correct functioning of the retina.

Beside the above mentioned general criticisms, more detailed comments are included below.

Lines 62-66: this sentence needs to be rephased as it is not the logic consequence of the previous one.

Lines 72-82: This sentence points to the fact that photoreceptors are the main oxygen consumers within the retina. However, since RGCs are also recognized as cells with high oxygen demand, the role of retinal cells other than photoreceptors on vascular abnormalities should be also discussed.

Lines 77-80: some words are missing and I have troubles to understand the sentence.

Lines 97-99: this sentence refers to tumors and should be adequately rephrased.

Chapter 4: when discussing the role of lipid metabolism as a source of energy and its role in retinal pathologies, the contribute of ketone bodies needs to be discussed as well as their possible role in slowing down DR progression.

Some sentences lack of references as, for instance, the sentence in lines 142-143.

Chapter 6: this chapter mainly refers to ROP. Additional discussion about the role of inflammatory processes in other ocular diseases is lacking although the availability of ample literature. In addition, the role of gliosis in retinal inflammatory processes should be also discussed.

Chapter 7: mechanisms underlying semaphorin action need to be discussed. Do semaphorins act through membrane receptors? Which downstream pathways are activated/inhibited by the different semaphorins in distinct cell types? Information about semaphorins in chapter 7 and in paragraph 8.7 should be put together.

As a general comment, the authors should make some distinction about the discussion on the role of angiogenesis in retinal development from that in retinal pathologies.

Lines 194-196: in which sense fenofibrate and FGF21 can be considered hormonal modulators? If the sense is that fenofibrate induces FGF21 expression that, in turn, increases adiponectin levels, it would be better to bring together paragraphs 8.1, 8.2 and 8.3.

If the aim of the authors is to “focus on the factors involved in neurovascular interactions which are potential therapeutic targets to decrease energy demand and/or to increase energy production for neovascular retinal disorders” they need to explain why additional mechanisms including for instance, AMPK or PGC-1 are not discussed in the present review.

Author Response

Reviewer 2

In the present review, the authors point to the role of the neurovascular unit in the retina focusing on the factors involved in neurovascular interactions as potential therapeutic targets to counteract neovascular retinal disorders. Retinal neurons, their supporting cells and the vascular network work in close coordination to integrate vascular flow with retinal metabolic activity. As a result of correct relationships, an appropriate environment contributes to correct metabolic need that is required for correct visual function.

Here, particular emphasis has been given to the critical role played by retinal photoreceptors in the neurovascular unit. In particular, the vascular demands by photoreceptors in hypoxic condition as in ROP cause increased levels of HIF and VEGF that determine neovessel proliferation affecting in turn retinal circuitry and function. In RP, photoreceptor loss causes a reduction of oxygen consumption that leads to reduced levels of HIF and VEGF causing, in turn, vessel attenuation that participates to retinal dysfunction. Additional mechanisms underlying neovessel formation have been also detailed by the authors as responsible of retinal neovascularization such as oxidative stress, inflammation, neuron-derived factors, but the list can be continued. The same applies to the list of potential therapeutic targets in retinal neurovascular disorders. Therefore, either the authors provide a more extensive revision of the literature or make some effort to allow the reader to better understanding the basic principles underlying the take home message of their review.

Response: we thank the reviewer for pointing this out. We agree with the reviewer that there are many potential factors involved in neurovascular interaction. For example, abnormal expression of numerous retinal miRNAs is related to retinal disorders such as AMD, DR, RP and retinoblastoma in both human and animal models [1]. In addition, miRNAs also regulates angiogenic factors and controls angiogenesis in vitro and in vivo [2-5]. In the current review, we focus on the major signals regulating oxygen and nutrients that mostly related to retinal metabolism, as well as some of the potential therapeutic targets that have been investigated in our group. This is now included in the revised manuscript.

In addition to include a more extensive revision on data on neurovascular network in the retina, the authors should also add some revision about the neuroretina and its dysfunction in relation to its interaction with the vascular network as also detailed in the title of the review. Although much is known on dysfunctional vascular network in retinal disorders, a more detailed revision on alterations in neuroretina circuitry coupled to diseased retinal vasculature needs to be provided. For instance, ROP models are known to exhibit a reduced number of astrocytes in the retina with no significant changes in the number of neuronal cells and the thickness of retinal layers suggesting that the reduced number of astrocytes leads to an impairment of cellular interactions between glial cells and neuronal and/or vascular cells characterizing ROP. In rodent models of RP, in contrast, loss of photoreceptors leads to major remodeling of retinal circuitry that leads to changes in the physiology of downstream retinal ganglion cells. Much discussion is needed about the reciprocal contribution of retinal vessels and retinal circuitry that, together, participate to a correct functioning of the retina.

Response: We thank the reviewer for the valuable suggestions. We now include the discussion of neuroretinal circuitry coupled to diseased retinal vasculature in the revised manuscript, also shown below.

Progressive photoreceptor (especially cone) loss in retinal degenerative diseases such as retinal detachment, AMD and RP, leads to extensive, phased changes in the remnant retinal circuitry – termed “retinal remodeling”, which is unavoidable at the cellular and molecular level in the inner retinal neurons and glial cells [6, 7]. First, photoreceptor stress and metabolic alterations in glia are initiated; second, glial remodeling occurs with the loss of photoreceptors; third, there is a neural, glial and vascular remodeling of the surviving retina. In RP patients, the severity of visual field loss is correlated with retinal vessel attenuation [8, 9]. In RP animal models, progressive loss of retinal blood vessels is also observed and a significant decrease in capillary density and capillary loop is found particularly in the deep (close to photoreceptors) but not in the superficial and intermediate capillary plexus [10, 11]. In ROP mouse models, hyperoxia causes reduced density of retinal astrocytes in the avascular zone and maintaining retinal astrocytes normalizes revascularization [12, 13]. Migrating astrocytes associate closely with the axons of retinal ganglion cells and subsequently direct vessel development in mice [14]. However, the thickness changes are only observed in the inner nuclear and plexiform layer but not photoreceptors in the mouse oxygen-induced retinopathy model at the time when maximum neovascularization occurs [15]. Therefore, photoreceptor stress-induced retinal remodeling may also control retinal vascular patterning possibly through modulating glial responses.

Beside the above mentioned general criticisms, more detailed comments are included below.

Lines 62-66: this sentence needs to be rephrased as it is not the logic consequence of the previous one.

Response: We thank the reviewer for pointing this out. We have now removed the discussion of capillary nonperfusion in the revised manuscript.

Lines 72-82: This sentence points to the fact that photoreceptors are the main oxygen consumers within the retina. However, since RGCs are also recognized as cells with high oxygen demand, the role of retinal cells other than photoreceptors on vascular abnormalities should be also discussed.

Response: Thanks for the comments. The discussion about roles of RGCs in vascular abnormalities is now included in the revised manuscript, also shown below.

In addition, retinal ganglion cells (RGCs) also play an important role in maintaining the normal structure and function of retinal blood vessels. Previous studies show that retinal vascular networks fail to develop in mice lacking RGCs [16]. RGCs also contribute to retinal blood vessels in experimental models of retinal regeneration induced by ischemia–reperfusion [17, 18]. These findings strongly suggest that RGCs play an important functional role in both physiological and pathological retinal angiogenesis [19, 20].

Lines 77-80: some words are missing and I have troubles to understand the sentence.

Response: We thank the reviewer for pointing this out. We have now revised the sentence as shown below.

In photoreceptor-degenerating mice exposed to oxygen in oxygen-induced retinopathy (OIR, modeling ROP), there are fewer preretinal vascular endothelial cell nuclei and reduced retinal VEGF expression level versus WT OIR controls [21].

Lines 97-99: this sentence refers to tumors and should be adequately rephrased.

Response: Thanks for the comments. This sentence was removed in the revision.

Chapter 4: when discussing the role of lipid metabolism as a source of energy and its role in retinal pathologies, the contribute of ketone bodies needs to be discussed as well as their possible role in slowing down DR progression.

Response: We thank for the reviewer’s valuable suggestion. We now include ketone bodies and their possible role in DR in the revised manuscript, also shown below.

Ketone bodies, mainly beta hydroxybutyrate and acetoacetate, are alternative energy sources in the fasting state [22]. Beta hydroxybutyrate activates G protein-coupled receptors GPR109A, which exhibits anti-inflammatory and neuroprotective effects [23]. Acetoacetate, as an endogenous agonist for GPR43, regulates energy expenditure and lipid metabolism in mice under fasting conditions [24]. Ketogenic diets are neuroprotective in some neuronal disorders including ischemic stroke [23], Parkinson’s disease [25] and Alzheimer’s disease [26]. Exogenous ketone bodies also show neuroprotective effects on retinas with enhanced antioxidative defenses [27]. RPE cells in vitro have high expression of key enzymes involved in ketogenesis and phagocytose photoreceptor OS to produce ketone bodies [28, 29]. RPE cells also metabolize ketone bodies for energy production [28]. High levels of ketone bodies are observed in patients with type 2 diabetes [30, 31]. Insulin treatment elevates serum ketone bodies in type 1 diabetes [32]. A recent conference report shows that beta hydroxybutyrate via GPR109A reduces NLRP3-derived inflammation and loss of GPR109A worsens retinal vascular pathology in streptozotocin-induced type 1 diabetic retinas [33]. In general, there might a protective role of ketone bodies in DR but the impacts of ketone bodies on diabetic retina need to be further investigated.

Some sentences lack of references as, for instance, the sentence in lines 142-143.

Response: We thank the reviewer for pointing this out. We now include the references as shown below.

Immune response and inflammation are associated with pathological angiogenesis as seen in ROP, AMD and DR [34-38].

Chapter 6: this chapter mainly refers to ROP. Additional discussion about the role of inflammatory processes in other ocular diseases is lacking although the availability of ample literature. In addition, the role of gliosis in retinal inflammatory processes should be also discussed.

Response: Thanks for the comments. Discussion of other diseases was added in the revision and the role of gliosis in retinal inflammatory processes is now included in the revised manuscript, also shown below.

Immune dysfunction and inflammation changes has been clinically and experimentally linked to development of choroidal neovascularization (CNV) [39-46], which affects ~10% of AMD patients, but it accounts for up to 90% of vision loss associated with AMD [47]. Immune cells play a key role in linking innate and adaptive immunity, primarily through antigen presentation and recruitment of adaptive immune cells. Choroidal circulation brings to the eye a large number of immune cells, specifically those of myeloid origin [48-50], which  play important roles in retinal and choroidal vascular pathology [44, 45, 51-54]. In response to chronic insults, such as those occurring in AMD, myeloid cells become activated and release inflammatory mediators (such as VEGF, matrix metalloproteinases, interleukins, and chemokines) that stimulate ocular NV [53, 55, 56]. A number of chemokines attract immune cells to invade and infiltrate ocular tissues, and furthermore stimulate the secretion of more trafficking molecules which influence immune cell migration [57]. Therefore, inflammatory mediators and the degree of myeloid cell activation and infiltration around CNV is critical to the innate inflammatory response that contributes to CNV onset and progression.

In addition, retinal gliosis had been reported to associate with ocular neovascularization, such as ROP [58] and DR [59].

Chapter 7: mechanisms underlying semaphorin action need to be discussed. Do semaphorins act through membrane receptors? Which downstream pathways are activated/inhibited by the different semaphorins in distinct cell types? Information about semaphorins in chapter 7 and in paragraph 8.7 should be put together.

Response: Thanks for the comments. We added the discussion about the mechanisms underlying semaphorin action and moved the discussion about class 3 semaphorins from chapter 7 to 8.7.

The action mechanisms underlying semaphorins through their receptors from different cell types in different disease conditions had been summarized previously [60-62]. Cerani, et.al reported that neuron-derived Semaphorin3A is an early inducer of vascular permeability in diabetic retinopathy via neuropilin-1[63]. Yang, et.al reported that Semaphorin-3C signals through Neuropilin-1 and PlexinD1 receptors to inhibit pathological angiogenesis [64]. Wu, et.al. reported that inhibition of Sema4D and its receptor Plexin B1 signaling alleviates vascular dysfunction in diabetic retinopathy [65]. Fukushima, et.al reported that Sema3E-PlexinD1 signaling selectively suppresses disoriented angiogenesis in ischemic retinopathy in mice [66]. These findings demonstrate that semaphorin family proteins are involved in neovascularization.

As a general comment, the authors should make some distinction about the discussion on the role of angiogenesis in retinal development from that in retinal pathologies.

Response: We appreciate the reviewer’s comment. We now include the discussion in Chapter 1 “Retinal neurovascular development” in the revised manuscript.

The timing and spatial arrangement of the retinal vasculature and retinal neurons suggest a neurovascular link. However, it is important to note that the signals involved in the neurovascular interaction and their impact might be different between developmental and pathological conditions. For example, retinal glia cell-derived VEGFA is important for physiological retinal angiogenesis, but high VEGFA expression in ischemic retinopathy induces pathological angiogenesis [67]. In addition, loss of WNT signaling leads to delayed hyaloid vessel regression and delayed retinal vascular growth, but over activation of WNT signaling leads to retinal neovascularization [68]. During the developmental stage, signals are tightly controlled for the proper formation of retinal vasculature. During disease conditions, the signals are uncontrolled and result in retinal vessel overgrowth.

Lines 194-196: in which sense fenofibrate and FGF21 can be considered hormonal modulators? If the sense is that fenofibrate induces FGF21 expression that, in turn, increases adiponectin levels, it would be better to bring together paragraphs 8.1, 8.2 and 8.3.

Response: We thank the reviewer for the suggestion. We now bring the three paragraphs together under section “3.1. Hormonal modulation”.

If the aim of the authors is to “focus on the factors involved in neurovascular interactions which are potential therapeutic targets to decrease energy demand and/or to increase energy production for neovascular retinal disorders” they need to explain why additional mechanisms including for instance, AMPK or PGC-1 are not discussed in the present review.

Response: We thank the reviewer for the suggestion. We now include the discussion of AMPK and PGC-1 under section “3.1. Hormonal modulation”.

Adenosine monophosphate-activated protein kinase (AMPK), a crucial cellular energy sensor, is activated by falling energy status [69]. Activation of AMPK via attenuating NF-κB activation protects retinal neurons against lipopolysaccharide-induced inflammation [70]. Activation of AMPK with metformin protects light-induced retinal degeneration with decreased oxidative stress and DNA damage, as well as increased mitochondrial energy production [71]. PGC-1α, which is stimulated by metabolic sensor AMPK and SIRT1, is a transcriptional coactivator of many genes involved in energy modulation and mitochondrial biogenesis [72, 73]. PGC-1α repression and mitochondrial dysfunction are observed in RPE derived from AMD donor eyes versus age-matched normal controls [74]. High-fat-diet induces AMD-like abnormalities in RPE and retinal morphology in mice with low levels of PGC-1α [75]. PGC-1α increases oxidative phosphorylation and fatty acid oxidation in RPE in vitro [76], as well as protects RPE cells of the aging retina against oxidative stress-induced degeneration in vivo [72]. Loss of PGC-1α in mice leads to rapid RPE dysfunction and promotes photoreceptor degeneration [77], as well as severe protects against a deterioration in retinal morphology and function with toxic light exposure [78]. PGC-1α also induces the expression of VEGFA in retinal cells and PGC-1α deficiency reduces early retinal vascular outgrowth, capillary density and number of arteries and veins as adults [79]. In a mouse model of ROP, PGC-1α expression is dramatically induced in the inner nuclear layer, and loss of PGC-1α inhibits retinal neovascularization by decreasing VEGFA levels [79].

Adiponectin induces AMPK activation by promoting the cytosolic localization of LKB1 and stimulating Ca2+ release from intracellular stores in muscle cells [80]. In hyperglycemia-associated ROP, adiponectin via AMPK increases photoreceptor metabolism and pro-angiogenic growth factor Pdgfb production, and in turn improves retinal vascular growth [81]. Endocrine FGF21 is also an AMPK activator either directly through FGFR1/β-klotho signaling or indirectly by stimulating the secretion of adiponectin and corticosteroids [82]. FGF21 controls mitochondrial function by activating AMPK-SIRT1-PGC-1α pathway in adipocytes [83].

  1. Andreeva, K.; Cooper, N. G., MicroRNAs in the Neural Retina. Int J Genomics 2014, 2014, 165897.
  2. Agrawal, S.; Chaqour, B., MicroRNA signature and function in retinal neovascularization. World J Biol Chem 2014, 5, (1), 1-11.
  3. Liu, C. H.; Wang, Z.; Huang, S.; Sun, Y.; Chen, J., MicroRNA-145 Regulates Pathological Retinal Angiogenesis by Suppression of TMOD3. Mol Ther Nucleic Acids 2019, 16, 335-347.
  4. Liu, C. H.; Wang, Z.; Sun, Y.; SanGiovanni, J. P.; Chen, J., Retinal expression of small non-coding RNAs in a murine model of proliferative retinopathy. Scientific reports 2016, 6, 33947.
  5. Liu, C. H.; Sun, Y.; Li, J.; Gong, Y.; Tian, K. T.; Evans, L. P.; Morss, P. C.; Fredrick, T. W.; Saba, N. J.; Chen, J., Endothelial microRNA-150 is an intrinsic suppressor of pathologic ocular neovascularization. Proceedings of the National Academy of Sciences of the United States of America 2015, 112, (39), 12163-8.
  6. Krishnamoorthy, V.; Cherukuri, P.; Poria, D.; Goel, M.; Dagar, S.; Dhingra, N. K., Retinal Remodeling: Concerns, Emerging Remedies and Future Prospects. Front Cell Neurosci 2016, 10, 38.
  7. Jones, B. W.; Pfeiffer, R. L.; Ferrell, W. D.; Watt, C. B.; Marmor, M.; Marc, R. E., Retinal remodeling in human retinitis pigmentosa. Experimental eye research 2016, 150, 149-65.
  8. Ma, Y.; Kawasaki, R.; Dobson, L. P.; Ruddle, J. B.; Kearns, L. S.; Wong, T. Y.; Mackey, D. A., Quantitative analysis of retinal vessel attenuation in eyes with retinitis pigmentosa. Investigative ophthalmology & visual science 2012, 53, (7), 4306-14.
  9. Nakagawa, S.; Oishi, A.; Ogino, K.; Makiyama, Y.; Kurimoto, M.; Yoshimura, N., Association of retinal vessel attenuation with visual function in eyes with retinitis pigmentosa. Clin Ophthalmol 2014, 8, 1487-93.
  10. Hanna, J.; Yucel, Y. H.; Zhou, X.; Mathieu, E.; Paczka-Giorgi, L. A.; Gupta, N., Progressive loss of retinal blood vessels in a live model of retinitis pigmentosa. Canadian journal of ophthalmology. Journal canadien d'ophtalmologie 2018, 53, (4), 391-401.
  11. Fernandez-Sanchez, L.; Esquiva, G.; Pinilla, I.; Lax, P.; Cuenca, N., Retinal Vascular Degeneration in the Transgenic P23H Rat Model of Retinitis Pigmentosa. Front Neuroanat 2018, 12, 55.
  12. Dorrell, M. I.; Aguilar, E.; Jacobson, R.; Trauger, S. A.; Friedlander, J.; Siuzdak, G.; Friedlander, M., Maintaining retinal astrocytes normalizes revascularization and prevents vascular pathology associated with oxygen-induced retinopathy. Glia 2010, 58, (1), 43-54.
  13. Bucher, F.; Stahl, A.; Agostini, H. T.; Martin, G., Hyperoxia causes reduced density of retinal astrocytes in the central avascular zone in the mouse model of oxygen-induced retinopathy. Mol Cell Neurosci 2013, 56, 225-33.
  14. O'Sullivan, M. L.; Punal, V. M.; Kerstein, P. C.; Brzezinski, J. A. t.; Glaser, T.; Wright, K. M.; Kay, J. N., Astrocytes follow ganglion cell axons to establish an angiogenic template during retinal development. Glia 2017, 65, (10), 1697-1716.
  15. Fu, Z.; Nian, S.; Li, S. Y.; Wong, D.; Chung, S. K.; Lo, A. C., Deficiency of aldose reductase attenuates inner retinal neuronal changes in a mouse model of retinopathy of prematurity. Graefe's archive for clinical and experimental ophthalmology = Albrecht von Graefes Archiv fur klinische und experimentelle Ophthalmologie 2015, 253, (9), 1503-13.
  16. Sapieha, P.; Sirinyan, M.; Hamel, D.; Zaniolo, K.; Joyal, J. S.; Cho, J. H.; Honore, J. C.; Kermorvant-Duchemin, E.; Varma, D. R.; Tremblay, S.; Leduc, M.; Rihakova, L.; Hardy, P.; Klein, W. H.; Mu, X.; Mamer, O.; Lachapelle, P.; Di Polo, A.; Beausejour, C.; Andelfinger, G.; Mitchell, G.; Sennlaub, F.; Chemtob, S., The succinate receptor GPR91 in neurons has a major role in retinal angiogenesis. Nature medicine 2008, 14, (10), 1067-76.
  17. Zheng, L.; Gong, B.; Hatala, D. A.; Kern, T. S., Retinal ischemia and reperfusion causes capillary degeneration: similarities to diabetes. Investigative ophthalmology & visual science 2007, 48, (1), 361-7.
  18. Ueda, K.; Nakahara, T.; Hoshino, M.; Mori, A.; Sakamoto, K.; Ishii, K., Retinal blood vessels are damaged in a rat model of NMDA-induced retinal degeneration. Neuroscience letters 2010, 485, (1), 55-9.
  19. Nakahara, T.; Mori, A.; Kurauchi, Y.; Sakamoto, K.; Ishii, K., Neurovascular interactions in the retina: physiological and pathological roles. J Pharmacol Sci 2013, 123, (2), 79-84.
  20. Sun, Y.; Liu, C. H.; Wang, Z.; Meng, S. S.; Burnim, S. B.; SanGiovanni, J. P.; Kamenecka, T. M.; Solt, L. A.; Chen, J., RORalpha modulates semaphorin 3E transcription and neurovascular interaction in pathological retinal angiogenesis. FASEB journal : official publication of the Federation of American Societies for Experimental Biology 2017, 31, (10), 4492-4502.
  21. Zhang, Q.; Zhang, Z. M., Oxygen-induced retinopathy in mice with retinal photoreceptor cell degeneration. Life sciences 2014, 102, (1), 28-35.
  22. Puchalska, P.; Crawford, P. A., Multi-dimensional Roles of Ketone Bodies in Fuel Metabolism, Signaling, and Therapeutics. Cell metabolism 2017, 25, (2), 262-284.
  23. Rahman, M.; Muhammad, S.; Khan, M. A.; Chen, H.; Ridder, D. A.; Muller-Fielitz, H.; Pokorna, B.; Vollbrandt, T.; Stolting, I.; Nadrowitz, R.; Okun, J. G.; Offermanns, S.; Schwaninger, M., The beta-hydroxybutyrate receptor HCA2 activates a neuroprotective subset of macrophages. Nature communications 2014, 5, 3944.
  24. Miyamoto, J.; Ohue-Kitano, R.; Mukouyama, H.; Nishida, A.; Watanabe, K.; Igarashi, M.; Irie, J.; Tsujimoto, G.; Satoh-Asahara, N.; Itoh, H.; Kimura, I., Ketone body receptor GPR43 regulates lipid metabolism under ketogenic conditions. Proceedings of the National Academy of Sciences of the United States of America 2019, 116, (47), 23813-23821.
  25. Tieu, K.; Perier, C.; Caspersen, C.; Teismann, P.; Wu, D. C.; Yan, S. D.; Naini, A.; Vila, M.; Jackson-Lewis, V.; Ramasamy, R.; Przedborski, S., D-beta-hydroxybutyrate rescues mitochondrial respiration and mitigates features of Parkinson disease. The Journal of clinical investigation 2003, 112, (6), 892-901.
  26. Yin, J. X.; Maalouf, M.; Han, P.; Zhao, M.; Gao, M.; Dharshaun, T.; Ryan, C.; Whitelegge, J.; Wu, J.; Eisenberg, D.; Reiman, E. M.; Schweizer, F. E.; Shi, J., Ketones block amyloid entry and improve cognition in an Alzheimer's model. Neurobiology of aging 2016, 39, 25-37.
  27. Izuta, Y.; Imada, T.; Hisamura, R.; Oonishi, E.; Nakamura, S.; Inagaki, E.; Ito, M.; Soga, T.; Tsubota, K., Ketone body 3-hydroxybutyrate mimics calorie restriction via the Nrf2 activator, fumarate, in the retina. Aging Cell 2018, 17, (1).
  28. Adijanto, J.; Du, J.; Moffat, C.; Seifert, E. L.; Hurle, J. B.; Philp, N. J., The retinal pigment epithelium utilizes fatty acids for ketogenesis. The Journal of biological chemistry 2014, 289, (30), 20570-82.
  29. Reyes-Reveles, J.; Dhingra, A.; Alexander, D.; Bragin, A.; Philp, N. J.; Boesze-Battaglia, K., Phagocytosis-dependent ketogenesis in retinal pigment epithelium. The Journal of biological chemistry 2017, 292, (19), 8038-8047.
  30. Avogaro, A.; Crepaldi, C.; Miola, M.; Maran, A.; Pengo, V.; Tiengo, A.; Del Prato, S., High blood ketone body concentration in type 2 non-insulin dependent diabetic patients. J Endocrinol Invest 1996, 19, (2), 99-105.
  31. Mahendran, Y.; Vangipurapu, J.; Cederberg, H.; Stancakova, A.; Pihlajamaki, J.; Soininen, P.; Kangas, A. J.; Paananen, J.; Civelek, M.; Saleem, N. K.; Pajukanta, P.; Lusis, A. J.; Bonnycastle, L. L.; Morken, M. A.; Collins, F. S.; Mohlke, K. L.; Boehnke, M.; Ala-Korpela, M.; Kuusisto, J.; Laakso, M., Association of ketone body levels with hyperglycemia and type 2 diabetes in 9,398 Finnish men. Diabetes 2013, 62, (10), 3618-26.
  32. Harano, Y.; Kosugi, K.; Hyosu, T.; Suzuki, M.; Hidaka, H.; Kashiwagi, A.; Uno, S.; Shigeta, Y., Ketone bodies as markers for type 1 (insulin-dependent) diabetes and their value in the monitoring of diabetic control. Diabetologia 1984, 26, (5), 343-8.
  33. Ollya Fromal, F. L., Matthew Kaufman, Manuela Bartoli, Pamela M Martin, Blockade of NLRP3 inflammasome activation in diabetic retina by the ketone metabolite beta-hydroxybutyrate is mediated by GPR109A In ARVO Annual Meeting, ARVO: Seattle, 2016; Vol. 57, p 5451.
  34. Rubsam, A.; Parikh, S.; Fort, P. E., Role of Inflammation in Diabetic Retinopathy. Int J Mol Sci 2018, 19, (4).
  35. Rathi, S.; Jalali, S.; Patnaik, S.; Shahulhameed, S.; Musada, G. R.; Balakrishnan, D.; Rani, P. K.; Kekunnaya, R.; Chhablani, P. P.; Swain, S.; Giri, L.; Chakrabarti, S.; Kaur, I., Abnormal Complement Activation and Inflammation in the Pathogenesis of Retinopathy of Prematurity. Front Immunol 2017, 8, 1868.
  36. Rivera, J. C.; Holm, M.; Austeng, D.; Morken, T. S.; Zhou, T. E.; Beaudry-Richard, A.; Sierra, E. M.; Dammann, O.; Chemtob, S., Retinopathy of prematurity: inflammation, choroidal degeneration, and novel promising therapeutic strategies. Journal of neuroinflammation 2017, 14, (1), 165.
  37. Kauppinen, A.; Paterno, J. J.; Blasiak, J.; Salminen, A.; Kaarniranta, K., Inflammation and its role in age-related macular degeneration. Cell Mol Life Sci 2016, 73, (9), 1765-86.
  38. Mansoor, N.; Wahid, F.; Azam, M.; Shah, K.; den Hollander, A. I.; Qamar, R.; Ayub, H., Molecular Mechanisms of Complement System Proteins and Matrix Metalloproteinases in the Pathogenesis of Age-Related Macular Degeneration. Curr Mol Med 2019, 19, (10), 705-718.
  39. Ambati, J.; Atkinson, J. P.; Gelfand, B. D., Immunology of age-related macular degeneration. Nature reviews. Immunology 2013, 13, (6), 438-51.
  40. Raychaudhuri, S.; Iartchouk, O.; Chin, K.; Tan, P. L.; Tai, A. K.; Ripke, S.; Gowrisankar, S.; Vemuri, S.; Montgomery, K.; Yu, Y.; Reynolds, R.; Zack, D. J.; Campochiaro, B.; Campochiaro, P.; Katsanis, N.; Daly, M. J.; Seddon, J. M., A rare penetrant mutation in CFH confers high risk of age-related macular degeneration. Nature genetics 2011, 43, (12), 1232-6.
  41. Doyle, S. L.; Campbell, M.; Ozaki, E.; Salomon, R. G.; Mori, A.; Kenna, P. F.; Farrar, G. J.; Kiang, A. S.; Humphries, M. M.; Lavelle, E. C.; O'Neill, L. A.; Hollyfield, J. G.; Humphries, P., NLRP3 has a protective role in age-related macular degeneration through the induction of IL-18 by drusen components. Nature medicine 2012, 18, (5), 791-8.
  42. Miller, J. W., Age-related macular degeneration revisited--piecing the puzzle: the LXIX Edward Jackson memorial lecture. American journal of ophthalmology 2013, 155, (1), 1-35 e13.
  43. Kumar, A.; Zhao, L.; Fariss, R. N.; McMenamin, P. G.; Wong, W. T., Vascular associations and dynamic process motility in perivascular myeloid cells of the mouse choroid: implications for function and senescent change. Investigative ophthalmology & visual science 2014, 55, (3), 1787-96.
  44. Skeie, J. M.; Mullins, R. F., Macrophages in neovascular age-related macular degeneration: friends or foes? Eye 2009, 23, (4), 747-55.
  45. Cherepanoff, S.; McMenamin, P.; Gillies, M. C.; Kettle, E.; Sarks, S. H., Bruch's membrane and choroidal macrophages in early and advanced age-related macular degeneration. Br J Ophthalmol 2010, 94, (7), 918-25.
  46. Chen, M.; Lechner, J.; Zhao, J.; Toth, L.; Hogg, R.; Silvestri, G.; Kissenpfennig, A.; Chakravarthy, U.; Xu, H., STAT3 Activation in Circulating Monocytes Contributes to Neovascular Age-Related Macular Degeneration. Curr Mol Med 2016, 16, (4), 412-23.
  47. Apte, R. S.; Richter, J.; Herndon, J.; Ferguson, T. A., Macrophages inhibit neovascularization in a murine model of age-related macular degeneration. PLoS Med 2006, 3, (8), e310.
  48. Yang, P.; de Vos, A. F.; Kijlstra, A., Macrophages and MHC class II positive cells in the choroid during endotoxin induced uveitis. Br J Ophthalmol 1997, 81, (5), 396-401.
  49. McMenamin, P. G., Dendritic cells and macrophages in the uveal tract of the normal mouse eye. Br J Ophthalmol 1999, 83, (5), 598-604.
  50. Forrester, J. V.; Xu, H.; Kuffova, L.; Dick, A. D.; McMenamin, P. G., Dendritic cell physiology and function in the eye. Immunological reviews 2010, 234, (1), 282-304.
  51. Karlstetter, M.; Scholz, R.; Rutar, M.; Wong, W. T.; Provis, J. M.; Langmann, T., Retinal microglia: just bystander or target for therapy? Progress in retinal and eye research 2015, 45, 30-57.
  52. Altmann, C.; Schmidt, M. H. H., The Role of Microglia in Diabetic Retinopathy: Inflammation, Microvasculature Defects and Neurodegeneration. Int J Mol Sci 2018, 19, (1).
  53. Grigsby, J. G.; Cardona, S. M.; Pouw, C. E.; Muniz, A.; Mendiola, A. S.; Tsin, A. T.; Allen, D. M.; Cardona, A. E., The role of microglia in diabetic retinopathy. J Ophthalmol 2014, 2014, 705783.
  54. Crespo-Garcia, S.; Reichhart, N.; Hernandez-Matas, C.; Zabulis, X.; Kociok, N.; Brockmann, C.; Joussen, A. M.; Strauss, O., In vivo analysis of the time and spatial activation pattern of microglia in the retina following laser-induced choroidal neovascularization. Experimental eye research 2015, 139, 13-21.
  55. Talia, D. M.; Deliyanti, D.; Agrotis, A.; Wilkinson-Berka, J. L., Inhibition of the Nuclear Receptor RORgamma and Interleukin-17A Suppresses Neovascular Retinopathy: Involvement of Immunocompetent Microglia. Arteriosclerosis, thrombosis, and vascular biology 2016, 36, (6), 1186-96.
  56. Vecino, E.; Rodriguez, F. D.; Ruzafa, N.; Pereiro, X.; Sharma, S. C., Glia-neuron interactions in the mammalian retina. Progress in retinal and eye research 2016, 51, 1-40.
  57. Schmid, M. C.; Varner, J. A., Myeloid cell trafficking and tumor angiogenesis. Cancer letters 2007, 250, (1), 1-8.
  58. Houston, S. K.; Bourne, T. D.; Lopes, M. B.; Ghazi, N. G., Bilateral massive retinal gliosis associated with retinopathy of prematurity. Archives of pathology & laboratory medicine 2009, 133, (8), 1242-5.
  59. Gu, L.; Xu, H.; Zhang, C.; Yang, Q.; Zhang, L.; Zhang, J., Time-dependent changes in hypoxia- and gliosis-related factors in experimental diabetic retinopathy. Eye 2019, 33, (4), 600-609.
  60. Alto, L. T.; Terman, J. R., Semaphorins and their Signaling Mechanisms. Methods in molecular biology 2017, 1493, 1-25.
  61. Toledano, S.; Nir-Zvi, I.; Engelman, R.; Kessler, O.; Neufeld, G., Class-3 Semaphorins and Their Receptors: Potent Multifunctional Modulators of Tumor Progression. Int J Mol Sci 2019, 20, (3).
  62. Franzolin, G.; Tamagnone, L., Semaphorin Signaling in Cancer-Associated Inflammation. Int J Mol Sci 2019, 20, (2).
  63. Cerani, A.; Tetreault, N.; Menard, C.; Lapalme, E.; Patel, C.; Sitaras, N.; Beaudoin, F.; Leboeuf, D.; De Guire, V.; Binet, F.; Dejda, A.; Rezende, F. A.; Miloudi, K.; Sapieha, P., Neuron-derived semaphorin 3A is an early inducer of vascular permeability in diabetic retinopathy via neuropilin-1. Cell metabolism 2013, 18, (4), 505-18.
  64. Yang, W. J.; Hu, J.; Uemura, A.; Tetzlaff, F.; Augustin, H. G.; Fischer, A., Semaphorin-3C signals through Neuropilin-1 and PlexinD1 receptors to inhibit pathological angiogenesis. EMBO molecular medicine 2015, 7, (10), 1267-84.
  65. Wu, J. H.; Li, Y. N.; Chen, A. Q.; Hong, C. D.; Zhang, C. L.; Wang, H. L.; Zhou, Y. F.; Li, P. C.; Wang, Y.; Mao, L.; Xia, Y. P.; He, Q. W.; Jin, H. J.; Yue, Z. Y.; Hu, B., Inhibition of Sema4D/PlexinB1 signaling alleviates vascular dysfunction in diabetic retinopathy. EMBO molecular medicine 2020, 12, (2), e10154.
  66. Fukushima, Y.; Okada, M.; Kataoka, H.; Hirashima, M.; Yoshida, Y.; Mann, F.; Gomi, F.; Nishida, K.; Nishikawa, S.; Uemura, A., Sema3E-PlexinD1 signaling selectively suppresses disoriented angiogenesis in ischemic retinopathy in mice. The Journal of clinical investigation 2011, 121, (5), 1974-85.
  67. Penn, J. S.; Madan, A.; Caldwell, R. B.; Bartoli, M.; Caldwell, R. W.; Hartnett, M. E., Vascular endothelial growth factor in eye disease. Progress in retinal and eye research 2008, 27, (4), 331-71.
  68. Wang, Z.; Liu, C. H.; Huang, S.; Chen, J., Wnt Signaling in vascular eye diseases. Progress in retinal and eye research 2019, 70, 110-133.
  69. Hardie, D. G.; Ross, F. A.; Hawley, S. A., AMPK: a nutrient and energy sensor that maintains energy homeostasis. Nat Rev Mol Cell Biol 2012, 13, (4), 251-62.
  70. Kamoshita, M.; Ozawa, Y.; Kubota, S.; Miyake, S.; Tsuda, C.; Nagai, N.; Yuki, K.; Shimmura, S.; Umezawa, K.; Tsubota, K., AMPK-NF-kappaB axis in the photoreceptor disorder during retinal inflammation. PloS one 2014, 9, (7), e103013.
  71. Xu, L.; Kong, L.; Wang, J.; Ash, J. D., Stimulation of AMPK prevents degeneration of photoreceptors and the retinal pigment epithelium. Proceedings of the National Academy of Sciences of the United States of America 2018, 115, (41), 10475-10480.
  72. Kaarniranta, K.; Kajdanek, J.; Morawiec, J.; Pawlowska, E.; Blasiak, J., PGC-1alpha Protects RPE Cells of the Aging Retina against Oxidative Stress-Induced Degeneration through the Regulation of Senescence and Mitochondrial Quality Control. The Significance for AMD Pathogenesis. Int J Mol Sci 2018, 19, (8).
  73. Canto, C.; Auwerx, J., PGC-1alpha, SIRT1 and AMPK, an energy sensing network that controls energy expenditure. Curr Opin Lipidol 2009, 20, (2), 98-105.
  74. Golestaneh, N.; Chu, Y.; Cheng, S. K.; Cao, H.; Poliakov, E.; Berinstein, D. M., Repressed SIRT1/PGC-1alpha pathway and mitochondrial disintegration in iPSC-derived RPE disease model of age-related macular degeneration. J Transl Med 2016, 14, (1), 344.
  75. Zhang, M.; Chu, Y.; Mowery, J.; Konkel, B.; Galli, S.; Theos, A. C.; Golestaneh, N., Pgc-1alpha repression and high-fat diet induce age-related macular degeneration-like phenotypes in mice. Dis Model Mech 2018, 11, (9).
  76. Iacovelli, J.; Rowe, G. C.; Khadka, A.; Diaz-Aguilar, D.; Spencer, C.; Arany, Z.; Saint-Geniez, M., PGC-1alpha Induces Human RPE Oxidative Metabolism and Antioxidant Capacity. Investigative ophthalmology & visual science 2016, 57, (3), 1038-51.
  77. Rosales, M. A. B.; Shu, D. Y.; Iacovelli, J.; Saint-Geniez, M., Loss of PGC-1alpha in RPE induces mesenchymal transition and promotes retinal degeneration. Life Sci Alliance 2019, 2, (3).
  78. Egger, A.; Samardzija, M.; Sothilingam, V.; Tanimoto, N.; Lange, C.; Salatino, S.; Fang, L.; Garcia-Garrido, M.; Beck, S.; Okoniewski, M. J.; Neutzner, A.; Seeliger, M. W.; Grimm, C.; Handschin, C., PGC-1alpha determines light damage susceptibility of the murine retina. PloS one 2012, 7, (2), e31272.
  79. Saint-Geniez, M.; Jiang, A.; Abend, S.; Liu, L.; Sweigard, H.; Connor, K. M.; Arany, Z., PGC-1alpha regulates normal and pathological angiogenesis in the retina. The American journal of pathology 2013, 182, (1), 255-65.
  80. Zhou, L.; Deepa, S. S.; Etzler, J. C.; Ryu, J.; Mao, X.; Fang, Q.; Liu, D. D.; Torres, J. M.; Jia, W.; Lechleiter, J. D.; Liu, F.; Dong, L. Q., Adiponectin activates AMP-activated protein kinase in muscle cells via APPL1/LKB1-dependent and phospholipase C/Ca2+/Ca2+/calmodulin-dependent protein kinase kinase-dependent pathways. The Journal of biological chemistry 2009, 284, (33), 22426-35.
  81. Fu, Z.; Lofqvist, C. A.; Liegl, R.; Wang, Z.; Sun, Y.; Gong, Y.; Liu, C. H.; Meng, S. S.; Burnim, S. B.; Arellano, I.; Chouinard, M. T.; Duran, R.; Poblete, A.; Cho, S. S.; Akula, J. D.; Kinter, M.; Ley, D.; Pupp, I. H.; Talukdar, S.; Hellstrom, A.; Smith, L. E., Photoreceptor glucose metabolism determines normal retinal vascular growth. EMBO Mol Med 2018, 10, (1), 76-90.
  82. Salminen, A.; Kauppinen, A.; Kaarniranta, K., FGF21 activates AMPK signaling: impact on metabolic regulation and the aging process. J Mol Med (Berl) 2017, 95, (2), 123-131.
  83. Chau, M. D.; Gao, J.; Yang, Q.; Wu, Z.; Gromada, J., Fibroblast growth factor 21 regulates energy metabolism by activating the AMPK-SIRT1-PGC-1alpha pathway. Proceedings of the National Academy of Sciences of the United States of America 2010, 107, (28), 12553-8.

Reviewer 3 Report

The authors focus on the factors involved in neurocascular interactions which are potential therapeutic targets to decrease energy demand and/or to increase energy production for neovascular retinal disorders. The authors reviewed more than a hundred of references and addressed the issues by specific topic. The chapter eight mentioned some potential therapeutic targets and methods. The chapter nine draw some conclusions and perspectives to show some promising field of studies. Here are some points the authors of this article can elaborate more.

Can increasing ATP production be used as potential therapy when patient already had neuro-dysfunction or in the aged patient? Figure 1 and figure 3 are nice schematic figures, but need to show who draw those figures? Where the figures are adapted? The last chapter (chapter 9) is short. Perhaps authors can summarize some ongoing clinical trial and/or ongoing funded research projects.

Author Response

Reviewer 3 The authors focus on the factors involved in neurocascular interactions which are potential therapeutic targets to decrease energy demand and/or to increase energy production for neovascular retinal disorders. The authors reviewed more than a hundred of references and addressed the issues by specific topic. The chapter eight mentioned some potential therapeutic targets and methods. The chapter nine draw some conclusions and perspectives to show some promising field of studies. Here are some points the authors of this article can elaborate more. 1. Can increasing ATP production be used as potential therapy when patient already had neuro-dysfunction or in the aged patient? Response: We thank the reviewer for the comment. We think that increasing ATP production would be protective in patients who already had neurodysfunction or in aged patients. For example, clinical trial of fenofibrate in type 2 diabetic patients demonstrates that fenofibrate decreases the disease progression in proliferative DR. Meanwhile, photobiomodulation in AMD patients also improves visual function. In ROP patients with neurovascular disorder, low adiponectin levels are associated with ROP progression and supplementation of APN via promoting photoreceptor glucose metabolism inhibits neurovascular dysfunction in experimental models. Taken together, we expect protective roles of increasing ATP production in retinal metabolic disorders. 2. Figure 1 and figure 3 are nice schematic figures, but need to show who draw those figures? Where the figures are adapted? Response: Schematics are drawn by Dr. Shuo Huang (Figure 1, 2) and Dr. Zhongjie Fu (Figure 3), Ophthalmology, BCH. The information is now included in the figure legend in revised manuscript. 3. The last chapter (chapter 9) is short. Perhaps authors can summarize some ongoing clinical trial and/or ongoing funded research projects. Response: We thank the reviewer’s suggestion. We now revise the last chapter as shown below. Neuronal metabolism and inflammatory responses control retinal vascular function. Hormonal modulation and photobiomodulation to increase ATP production, or inhibition of the visual cycle may improve retinal function. Furthermore, controlling photoreceptor inflammatory responses by modulating c-Fos and SOCS3 pathways, as well as regulating neuron-derived Class 3 SEMAs to inhibit the pathologic vessel formation is also a promising field of study. To date, fenofibrate treatment in type 2 diabetic patients reduces proliferative DR. Photobiomodulation in AMD patients also improves visual function. Long-acting FGF21 improves circulating lipid profiles and decreases body weight in type 2 diabetic patients. The investigation of FGF21 on retinal metabolic disorders are in the pre-clinical phase but clinical trials of long-acting FGF21 in patients with non-alcoholic steatohepatitis is ongoing. The development of safe and effective visual cycle inhibitors for retinal degeneration and proliferative retinopathies, and further exploration of modulating retinal inflammatory responses by c-Fos and SOCS3 in proliferative retinopathies are currently in process.

Round 2

Reviewer 2 Report

the manuscript can be accepted in its revised version